# GMAIL: Generative Modality Alignment for generated Image Learning

**Shentong Mo** [1 2]   **Sukmin Yun** [3]

## Abstract

Generative models have made it possible to synthesize highly realistic images, potentially providing an abundant data source for training machine learning models. Despite the advantages of these synthesizable data sources, the indiscriminate use of generated images as real images for training can even cause mode collapse due to modality discrepancies between real and synthetic domains. In this paper, we propose a novel framework for discriminative use of generated images, coined *GMAIL*, that explicitly treats generated images as a separate modality from real images. Instead of indiscriminately replacing real images with generated ones in the pixel space, our approach bridges the two distinct modalities in the same latent space through a multi-modal learning approach. To be specific, we first fine-tune a model exclusively on generated images using a cross-modality alignment loss and then employ this aligned model to further train various vision-language models with generated images. By aligning the two modalities, our approach effectively leverages the benefits of recent advances in generative models, thereby boosting the effectiveness of generated image learning across a range of vision-language tasks. Our framework can be easily incorporated with various vision-language models, and we demonstrate its efficacy throughout extensive experiments. For example, our framework significantly improves performance on image captioning, zero-shot image retrieval, zero-shot image classification, and long caption retrieval tasks. It also shows positive generated data scaling trends and notable enhancements in the captioning performance of the large multimodal model, LLaVA.

[1]Department of Machine Learning, CMU, USA [2]Department of Machine Learning, MBZUAI, UAE [3]Department of Artificial Intelligence, Hanyang University ERICA, South Korea. Correspondence to: Sukmin Yun <sukminyun@hanyang.ac.kr>.

*Proceedings of the 42nd International Conference on Machine Learning*, Vancouver, Canada. PMLR 267, 2025. Copyright 2025 by the author(s).

## 1. Introduction

Generative models, such as GANs (Goodfellow et al., 2020; Chen et al., 2016) and diffusion models (Song et al., 2021a; Dhariwal & Nichol, 2021; Rombach et al., 2022), have revolutionized the field of computer vision by enabling the synthesis of highly realistic images. These generated images offer a rich and scalable source of data, which can significantly augment training datasets, enhance data diversity, and reduce the dependency on costly real-world data collection. However, despite their potential, incorporating generated images directly into training pipelines poses substantial challenges due to inherent modality discrepancies between generated and real images. This misalignment often leads to a phenomenon known as mode collapse (LeCun, 2022), where the model's performance severely deteriorates due to an over-reliance on generated content that fails to generalize well to real-world scenarios. To address this, it is essential to solve the generated-to-real (Gen-Real) modality discrepancy problem first.

Existing approaches (Tian et al., 2024) typically integrate generated images into the training process without adequately addressing the modality gap between generated and real images. The resulting models are prone to overfitting the peculiarities of synthetic data, which negatively impacts performance across various downstream tasks, particularly when the model encounters real-world data. The primary source of this collapse lies in the failure to recognize that generated images, despite their realism, represent a distinct data modality that deviates from real images in subtle but significant ways. Addressing this modality gap is crucial to harnessing the full potential of generated data while maintaining robust performance on real-world tasks.

The challenge of using generated images stems from the fundamental differences between generated and real-world data distributions. Even when generated images appear visually convincing, they often contain subtle artifacts, biases, or domain-specific noise introduced during the generation process. These discrepancies are not just visual but can also affect higher-level semantic representations, resulting in a misalignment in the feature space that can propagate through the training pipeline. Furthermore, generative models may inadvertently capture and amplify biases present in their training data, leading to synthetic images that devi-

ate in unexpected ways from real-world distributions. This modality gap poses significant challenges for downstream tasks, where models trained on misaligned data struggle with overfitting to generated features, reduced robustness, and degraded performance when applied to real images. Bridging this gap is critical to leveraging the strengths of generative models while avoiding pitfalls that compromise model reliability.

To tackle this challenge, we introduce a novel framework for **G**enerative **M**odality **A**lignment for generated **I**mage **L**earning, namely *GMAIL*, that explicitly treats generated images as a separate modality from real images. Unlike conventional methods that mix generated and real data indiscriminately, our approach bridges the two distinct modalities in the latent space by embedding generated images alongside real images having the same descriptions. Specifically, we fine-tune a model exclusively on generated images using a cross-modality alignment loss while keeping the pre-trained model for real images unchanged. This allows for explicit and adaptive alignment between the two modalities, enabling us to utilize the aligned model for training various vision-language models (Radford et al., 2021; Liu et al., 2023; Zhang et al., 2024) with highly realistic generated images. Thereby, we fully exploit the advantages of recent advances in generative models (Rombach et al., 2022), enhancing the performance of generated image training across various vision-language tasks.

Through the extensive experiments across a wide range of vision-language tasks, we demonstrate the effectiveness of our framework by incorporating it with various vision-language models such as LLaVA (Liu et al., 2023). For example, our approach enhances image captioning on COCO (Lin et al., 2014), zero-shot image retrieval on COCO (Lin et al., 2014) and Flickr30k (Young et al., 2014), zero-shot image classification across eight widely used datasets, and long caption retrieval on ShareGPT4V (Chen et al., 2024). Furthermore, we observe positive generated data scaling trends in our framework across diverse datasets such as COCO (Lin et al., 2014), CC3M (Sharma et al., 2018), and CC12M (Changpinyo et al., 2021), highlighting the scalability of our method. Notably, our approach also improves the captioning performance of the recent large multimodal model, LLaVA (Liu et al., 2023), demonstrating its broad compatibility.

Our main contributions are summarized as:

- We introduce a novel framework for discriminative use of generated images, explicitly treating them as a distinct modality and aligning them with real images within the same latent space. It enables researchers to exploit highly realistic generated images effectively.

- We demonstrate the effectiveness of our framework

through extensive experiments on a diverse set of vision-language benchmarks, including image captioning, zero-shot image retrieval, and zero-shot image classification, and further validate its compatibility with the recent large multimodal model, LLaVA.

- We explore the generated data scaling trend of our framework using large-scale generated datasets, demonstrating that our approach consistently improves as the volume of training data increases.

## 2. Related Work

**Diffusion Models.** Diffusion models (Ho et al., 2020; Song et al., 2021b;a) have emerged as a powerful class of generative models, capable of producing high-quality images that closely mimic the distribution of real-world images. Prominent examples include Stable-Diffusion (Rombach et al., 2022), DreamBooth (Ruiz et al., 2023; 2024), and the DALL-E series (Ramesh et al., 2021; 2022; Betker et al., 2023), which have demonstrated remarkable success in generating diverse and complex images from textual descriptions. These models leverage advanced diffusion processes to iteratively refine images from noise, capturing intricate details and generating visually convincing outputs that can closely resemble real-world imagery. Our work utilizes the power of diffusion models to generate images, offering an innovative and cost-effective source of training data derived from textual descriptions. By aligning these generated images with real image modalities through our GMAIL framework, we bridge the gap between synthetic image generation and practical machine learning applications, addressing the challenges of modality discrepancies. This application of diffusion models represents a novel contribution to the field, as it not only enhances training efficiency but also expands the use of generative models beyond mere content creation, embedding them directly into the model training process to improve real-world performance.

**Generated Image Learning.** Generated image learning has gained traction as researchers explore the potential of synthetic data to augment traditional training paradigms. Syn-CLR (Tian et al., 2024) proposed a self-supervised framework that employs synthetic data to pre-train visual representations, showing that models trained on generated data achieve competitive results compared to those trained on real data. However, a critical challenge in this domain is the issue of mode collapse, where the over-reliance on synthetic data without proper alignment leads to performance degradation when models are applied to real-world tasks. Recent work (Shumailov et al., 2024) highlights the inherent risks of training models on recursively generated data, emphasizing that models can inherit and amplify errors in synthetic data, ultimately compromising their ability to generalize. Our research directly addresses these challenges by propos-

ing a novel strategy that treats generated images as a distinct modality and aligns them with real images in the same latent space. This approach not only mitigates the risk of collapse but also enhances the robustness by embedding generated images within the same latent space as real images.

Meanwhile, there also exist attempts (Zhang et al., 2021; Ye et al., 2024; Wu et al., 2023) at generating high-quality labeled datasets. While prior works have primarily focused on improving image segmentation tasks by generating mask labels, we instead aim to bridge modality gaps when using synthetic data to further train vision–language models, mitigating over-reliance on the synthetic domain.

**Vision-Language Models.** Vision-language models, such as CLIP (Radford et al., 2021), have revolutionized cross-modal understanding by learning joint representations of images and text through contrastive learning objectives. While these models excel at leveraging large-scale real-world data, they often struggle when trained on generated images due to the modality gap. To overcome this, recent methods have explored various alignment techniques to improve cross-modal performance. For example, Long-CLIP (Zhang et al., 2024) extended CLIP by integrating longer captions, improving its ability to handle more descriptive texts. Similarly, LLaVA (Liu et al., 2023) has demonstrated the potential for vision-language models to handle multimodal tasks like visual question answering and captioning by leveraging large-scale data. Our work builds on these foundational efforts by introducing an explicit Gen-Real alignment framework that enhances the adaptability of vision-language models when using generated data. By embedding generated images within the same latent space as real images and training the alignment, our approach directly addresses the modality discrepancies that limit model performance, offering a scalable solution that significantly boosts cross-modal learning across diverse vision-language tasks, including image captioning, zero-shot retrieval, and classification.

## 3. Method

In this section, we describe our proposed **G**enerative **M**odality **A**lignment for generated **I**mage **L**earning (*GMAIL*) framework, which tackles the challenge of training on generated images while ensuring robust performance during inference on real-world data, as illustrated in Figure 1. Our approach introduces two key components: (1) a Gen-CLIP flow on training and inference that handles generated and real images as separate modalities, and (2) an explicit alignment strategy with vision-language models to facilitate better integration with large language models (LLMs) such as CLIPCap (Mokady et al., 2021), LLaVA (Liu et al., 2023), and Llama3 (Meta, 2024). In this part, we detail the problem setup, the key components of our framework, and the alignment strategy used to enhance the performance of

models trained on both generated and real data.

### 3.1. Preliminaries

In this subsection, we introduce the problem setup and notations, followed by an overview of the contrastive language-image pre-training methodology that forms the foundation of our approach.

**Problem Setup and Notations.** Let $\mathcal{D}_r = \{(x_r, y_r)\}$ represent a dataset of real images with corresponding labels or annotations, and $\mathcal{D}_g = \{(x_g, y_g)\}$ denote a dataset of generated images synthesized by generative models, such as GANs or diffusion models. Our objective is to train a model $f(\cdot)$ that performs well across a broad set of downstream tasks, utilizing both $\mathcal{D}_r$ and $\mathcal{D}_g$, while mitigating the risk of mode collapse caused by the inherent modality gap between $\mathcal{D}_r$ and $\mathcal{D}_g$. To formally define the alignment process, we introduce two models: a base model $f_r$, pre-trained on real images, and a fine-tuned model $f_g$, trained specifically on generated images. The primary goal of our framework is to align $f_g$ with $f_r$, ensuring that the feature representations of generated images are semantically consistent with those of real images. This alignment facilitates a unified understanding of both modalities, allowing the model to generalize across real data during inference.

**Contrastive Language-Image Pre-training.** Our framework builds on the foundation of Contrastive Language-Image Pre-Training (CLIP) (Radford et al., 2021), which learns joint embeddings for images and textual descriptions. CLIP leverages a contrastive loss that brings the embeddings of paired images and texts closer, while pushing apart the embeddings of unpaired ones, fostering cross-modal alignment. However, traditional CLIP training does not explicitly address the discrepancy between generated and real images, often leading to performance degradation when integrating generated data directly. To extend CLIP to handle generated images as a distinct modality, we propose a modified training objective that incorporates contrastive learning not only between real images and text but also between generated images and text. This treats generated and real images independently, preserving the unique characteristics of each modality during training.

### 3.2. Gen-CLIP Flow: Training on Generated Images

The first key component of our method is the *Gen-CLIP* flow, which focuses on training the model on generated images while treating them as a distinct modality. Unlike traditional approaches that mix generated and real images indiscriminately, we handle generated images separately to prevent the model from overfitting to the peculiarities of synthetic data. In the *Gen-CLIP* flow, we fine-tune a pre-trained CLIP model (*i.e.,* image encoder of $f_r$) using generated images, paired with the same textual descriptions

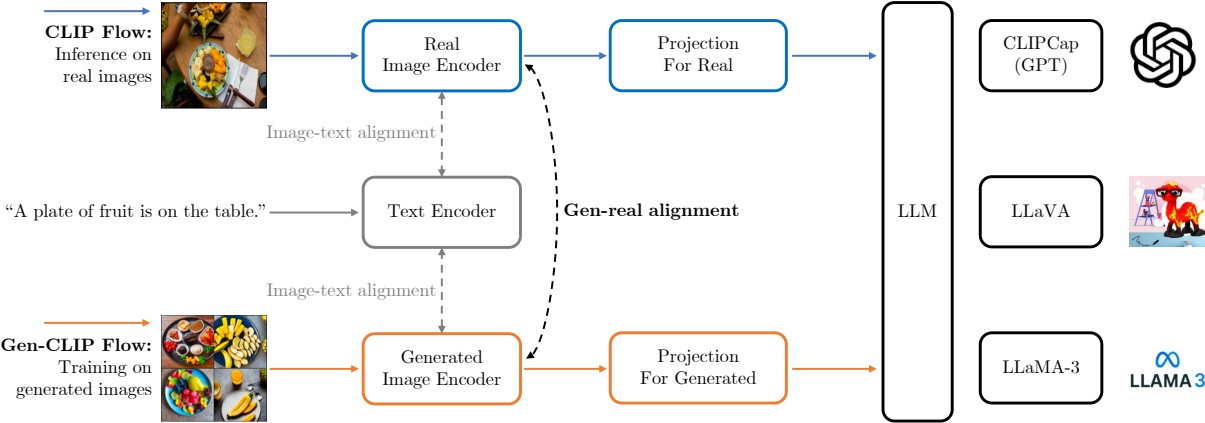

*Figure 1.* **Illustration of the proposed framework for vision-language tuning with Gen-Real alignment from diffusion models.** We propose a method that explicitly aligns a CLIP trained on real images with another CLIP model trained on generated images and then leverages the aligned CLIP to train state-of-the-art vision-language models on generated images (*i.e.,* Gen-CLIP Flow). For inference with real images, the original CLIP is used to process real images, thereby avoiding discrepancies between real and generated modalities.

used for real images. During fine-tuning, we employ a cross-modality alignment loss to minimize the feature space discrepancy between generated and real images (see Eq. 1). This contrastive alignment loss encourages the model $f_g$ to learn representations that place generated and real images with the same descriptions close to each other in the latent space, while maintaining their distinct modality-specific characteristics. To maintain computational efficiency and prevent catastrophic forgetting of real image representations, we apply Low-Rank Adaptation (LoRA) (Hu et al., 2022) during fine-tuning. LoRA introduces lightweight, efficient updates to the model, ensuring that the alignment process does not degrade the model's ability to generalize across different data modalities.

**CLIP Flow: Inference on Real Images.** In the inference phase with real images, the model $f_g$ fine-tuned on generated images in the *Gen-CLIP* flow can be deployed to process real images through the original image encoder of $f_r$ without further fine-tuning. By keeping the pre-trained CLIP model for real images unchanged during the generated image training process, we ensure that the learned representations from the generated data remain aligned with real data. The *CLIP* flow leverages these aligned representations for inference on real images, allowing the model to generalize well to real-world data without suffering from the typical mode collapse associated with over-reliance on generated content. This dual-model structure allows the model to benefit from the complementary strengths of both real and generated images, ensuring that it performs robustly during real-world deployment while still benefiting from the scalability of generated training data. Note that the encoder fine-tuned with the LoRA and the projection for real is used on real images during inference time.

## 3.3. Alignment with Vision-Language Models

Our alignment strategy is designed to enhance the integration of generated data into vision-language models, particularly large language models (LLMs) such as CLIP-Cap (Mokady et al., 2021), LLaVA (Liu et al., 2023), and Llama3 (Meta, 2024). This extension of GMAIL ensures that generated images can be utilized effectively within these models for tasks such as image captioning, retrieval, and long-form question answering.

**Gen-Real Alignment.** The key to our framework is the cross-modality alignment loss, which ensures that generated images are embedded within the same latent space as real images, while maintaining their distinct characteristics. The alignment loss is formulated as:

$$\mathcal{L}_{align} = -\frac{1}{|\mathcal{B}|} \sum_{(x_g, x_r) \in \mathcal{B}}$$
$$\log \frac{\exp(\text{sim}(f_g(x_g), f_r(x_r))/\tau)}{\sum_{x'_r \in \mathcal{B}} \exp(\text{sim}(f_g(x_g), f_r(x'_r))/\tau)}, \quad (1)$$

where $x_g$ and $x_r$ represent generated and real images, $f_g$ and $f_r$ are their corresponding image representations, $\text{sim}(\cdot, \cdot)$ denotes cosine similarity between embeddings, and $\tau$ is a temperature parameter. This loss encourages generated images to be aligned with their real counterparts, facilitating effective transfer of knowledge across both modalities.

**CLIPCap (Mokady et al., 2021)** combines CLIP's image embeddings with a transformer-based language model to generate captions from images. By aligning generated images with real image embeddings, we ensure that CLIPCap can generate high-quality captions from both real and generated data. Fine-tuning CLIPCap with our alignment frame-

work allows the model to handle both modalities effectively, resulting in enhanced performance on image captioning.

**LLaVA (Liu et al., 2023) & Llama3 (Meta, 2024)** are advanced multimodal models designed to perform vision-language tasks. To align generated images with these models, we first fine-tune the vision-language models using our GMAIL strategy to ensure that representations from generated data are aligned with real data. The aligned vision representations are then integrated with the LLMs, allowing the models to handle complex vision-language tasks such as long captioning and retrieval more effectively. This alignment enhances the robustness and flexibility of LLaVA and Llama3 in both real and generated images.

Our framework is designed to scale effectively with larger datasets, as evidenced by the performance improvements observed on large-scale datasets such as CC3M (Sharma et al., 2018) and CC12M (Changpinyo et al., 2021). The alignment strategy ensures that as the volume of generated training data increases, the model continues to generalize effectively to real-world data. This scalability demonstrates the potential of GMAIL as a cost-effective solution for training robust vision-language models using synthetic data.

## 4. Experiments

In this section, we provide the experimental setup, evaluation metrics, and comparative analysis conducted to validate the effectiveness of our method. Through rigorous experimentation on a diverse set of datasets, we assess our model on image captioning, zero-shot image retrieval, and zero-shot image classification tasks, comparing it against existing benchmarks to highlight our contributions.

### 4.1. Experimental Setup

**Datasets.** Our experiments leverage a comprehensive collection of datasets to evaluate the versatility and effectiveness of our proposed Gen-Real alignment framework. We focus on a diverse set of tasks, including image captioning, zero-shot image retrieval, and zero-shot image classification, ensuring broad coverage across various domains. Please refer to Appendix Section A for the detailed dataset settings.

**Evaluation Metrics.** To comprehensively evaluate our framework, we employ task-specific metrics tailored to image captioning, zero-shot image retrieval, and zero-shot image classification: **Image Captioning**: Performance is assessed using standard metrics such as BLEU@4 (B@4) (Papineni et al., 2002), METEOR (Denkowski & Lavie, 2014), CIDEr (Vedantam et al., 2014), SPICE (Anderson et al., 2016), ROUGE-L (Lin & Och, 2004), and Word Mover's Distance (WMD) (Kusner et al., 2015). These metrics evaluate the quality and semantic accuracy of generated captions compared to ground truth. **Zero-Shot Image Retrieval**:

We measure both image-to-text and text-to-image retrieval capabilities using Recall@1, Recall@5, and Recall@10. These metrics assess the model's ability to correctly retrieve relevant items based on the provided query, highlighting its cross-modal understanding. **Zero-Shot Image Classification**: The classification performance on unseen categories is evaluated using top-1 accuracy, reflecting the model's generalization ability to new classes without prior training on those specific categories.

**Implementation.** For image captioning, we adhere to the implementation strategy of ClipCap (Mokady et al., 2021), which combines CLIP with a text generation model to produce descriptive captions for images. ClipCap uses CLIP's image embeddings as input to a transformer-based captioning model, enabling the generation of semantically accurate and contextually rich captions for both real and generated images. For zero-shot evaluation on both retrieval and image classification tasks, we follow the setup detailed in the original CLIP (Radford et al., 2021) paper. This setup emphasizes the model's ability to generalize across unseen data by using natural language prompts to guide image classification and retrieval, leveraging the contrastive training between images and textual descriptions without explicit fine-tuning on target datasets. We adopt Stable Diffusion v2 (Rombach et al., 2022) to generate synthetic images using captions from the COCO (Lin et al., 2014) train2014 set. Stable Diffusion provides high-quality image synthesis, enabling us to produce generated images that are both visually realistic and semantically aligned with the training captions, serving as the generated modality in our alignment framework. During fine-tuning, we use a rank of 4 in Low-Rank Adaptation (LoRA) to adjust the model parameters specifically for generated images, ensuring that the adaptation remains efficient and computationally manageable. For optimization, we use the AdamW optimizer with a learning rate of $1 \times 10^{-4}$ and weight decay of $0.01$. We employ a cosine annealing schedule with warm restarts to dynamically adjust the learning rate, enhancing convergence stability across training phases. Batch normalization and gradient clipping are applied to prevent exploding gradients and ensure smooth training dynamics.

### 4.2. Comparison to prior work

**Image Captioning.** We compare our model's performance on the COCO dataset against prior commonly-used baselines, including ClipCap (Mokady et al., 2021), LLaVA (Liu et al., 2023), and LLAMA-3 (Meta, 2024). The results, detailed in Table 1, demonstrate significant improvements across all evaluated metrics, underscoring the efficacy of our GMAIL approach when combined with synthetic images and LoRA optimization. For ClipCap, the proposed ClipCap + GMAIL configuration achieves 38.12 B@4, 31.67 METEOR, 119.53 CIDEr, 23.75 SPICE, 56.27 ROUGE-L,

*Table 1.* **Image captioning.** We perform fine-tuning on pre-trained ClipCap, IFCap, LLaVA, and LLaMA-3 for image captioning on COCO. We report the standard metrics to evaluate the quality of generated captions. The best results are indicated in **bold**.

| Method | B@4 (↑) | METEOR(↑) | CIDEr (↑) | SPICE (↑) | ROUGE-L (↑) | WMD (↑) |
|---|---|---|---|---|---|---|
| ClipCap (Mokady et al., 2021) | 32.15 | 27.10 | 108.35 | 20.12 | – | – |
| ClipCap + GMAIL (ours) | **38.12** | **31.67** | **119.53** | **23.75** | **56.27** | **62.16** |
| IFCap (Lee et al., 2024) | 33.25 | 28.60 | 115.27 | 21.58 | 51.35 | 56.72 |
| IFCap + GAMIL (ours) | **39.32** | **32.07** | **127.86** | **23.98** | **59.83** | **63.51** |
| LLaVA (Liu et al., 2023) | 39.67 | 32.38 | 134.29 | 24.17 | 61.36 | 65.78 |
| LLaVA + GMAIL (ours) | **43.26** | **34.89** | **146.38** | **27.23** | **65.25** | **71.39** |
| Llama3 (Meta, 2024) | 47.36 | 35.21 | 158.13 | 28.35 | 68.32 | 75.13 |
| Llama3 + GMAIL (ours) | **50.21** | **38.59** | **168.53** | **32.58** | **73.29** | **80.25** |

*Table 2.* **Zero-shot image retrieval on COCO.** We perform zero-shot retrieval on pre-trained CLIP models for image retrieval on the COCO benchmark. We report the image-to-text and text-to-image Recall@1,5,10 metrics to evaluate the quality of retrieved images.

| Method | Image-to-Text | | | Text-to-Image | | |
|---|---|---|---|---|---|---|
| | R@1 (↑) | R@5 (↑) | R@10 (↑) | R@1 (↑) | R@5 (↑) | R@10 (↑) |
| CLIP (Radford et al., 2021) | 51.8 | 76.8 | 84.3 | 32.7 | 57.7 | 68.2 |
| CLIP + GMAIL (ours) | **56.8** | **80.1** | **87.2** | **37.5** | **62.7** | **73.2** |
| Long-CLIP (Zhang et al., 2024) | 57.2 | 80.8 | 87.8 | 40.4 | 65.9 | 75.7 |
| Long-CLIP + GMAIL (ours) | **62.3** | **84.1** | **91.2** | **45.6** | **69.8** | **79.5** |

and 62.16 WMD, significantly outperforming the baseline ClipCap and the ClipCap + LoRA setup. Specifically, our GMAIL approach boosts the original ClipCap (Mokady et al., 2021) by 5.97 B@4, 4.57 METEOR, 11.18 CIDEr, and 3.63 SPICE. These results highlight the advantages of aligning generated and real images within a unified semantic space, allowing for enhanced image captioning performance. Similarly, when applied to LLAMA-3, our LLAMA-3 + GMAIL model reaches 50.21 B@4, 38.59 METEOR, 168.53 CIDEr, 32.58 SPICE, 73.29 ROUGE-L, and 80.25 WMD, demonstrating notable improvements over both the baseline and the LoRA fine-tuning strategy. Compared to LLAMA-3 alone, GMAIL achieves gains of 2.85 B@4, 2.46 METEOR, 10.35 CIDEr, and 4.30 SPICE, establishing our approach as a robust technique for enhancing models through Gen-Real alignment. The substantial gains observed across both model architectures confirm the effectiveness of our GMAIL framework. By fine-tuning with generated images while maintaining alignment with real image modalities, our method effectively bridges the modality gap, resulting in better understanding and generation of descriptive captions aligned with real-world data.

**Zero-shot Image Retrieval.** The comparative results in Tables 2 and 3 highlight our model's superior recall rates, showcasing its robustness in understanding and associating visual and textual data. Our method is evaluated on two benchmarks: COCO and Flickr30k, using both image-to-text and text-to-image retrieval tasks, demonstrating significant improvements over prior baselines. On the COCO dataset, our approach, CLIP + GMAIL, achieves 56.8 R@1, 80.1 R@5, and 87.2 R@10 for image-to-text retrieval, outperforming the original CLIP (Radford et al., 2021) trained

on real images by 5.0 R@1, 3.3 R@5, and 2.9 R@10. For text-to-image retrieval, CLIP + GMAIL scores 37.5 R@1, 62.7 R@5, and 73.2 R@10, demonstrating gains of 4.8 R@1, 5.0 R@5, and 5.0 R@10 compared to the baseline CLIP. These improvements validate the effectiveness of our alignment strategy in bridging the gap between generated and real image modalities, enhancing zero-shot retrieval capabilities. Similarly, when applied to the Long-CLIP architecture (Zhang et al., 2024), our Long-CLIP + GMAIL configuration further boosts performance, achieving 62.3 R@1, 84.1 R@5, and 91.2 R@10 on image-to-text retrieval, and 45.6 R@1, 69.8 R@5, and 79.5 R@10 on text-to-image retrieval. This demonstrates that GMAIL consistently enhances model performance across different backbone architectures by facilitating better alignment of generated images with real-world data. On the Flickr30k dataset, our CLIP + GMAIL model achieves 47.1 R@1, 71.2 R@5, and 79.6 R@10 for image-to-text retrieval, outperforming CLIP by 3.0 R@1, 3.2 R@5, and 2.6 R@10. In text-to-image retrieval, the model scores 39.3 R@1, 61.5 R@5, and 71.8 R@10, with respective gains of 14.6 R@1, 16.0 R@5, and 17.2 R@10 over CLIP. These results validate the robustness of our approach in learning meaningful representations from generated images for zero-shot retrieval on real images.

**Zero-shot Image Classification.** We evaluate the zero-shot classification performance of our model across eight diverse benchmarks, including DTD, Stanford Cars, SUN397, Food 101, Aircraft, Oxford Pets, Caltech 101, and ImageNet 1K. As shown in Table 4, our model consistently achieves top-1 accuracy surpassing previous approaches, validating the advantage of leveraging generated images through our framework for enhancing zero-shot learning capabilities.

*Table 3.* **Zero-shot image retrieval on Flickr30k.** We perform zero-shot retrieval on pre-trained CLIP models for image retrieval on the Flickr30k benchmark. We report the image-to-text and text-to-image Recall@1,5,10 metrics to evaluate the quality of retrieved images.

| Method | Image-to-Text | | | Text-to-Image | | |
|---|---|---|---|---|---|---|
| | R@1 (↑) | R@5 (↑) | R@10 (↑) | R@1 (↑) | R@5 (↑) | R@10 (↑) |
| CLIP (Radford et al., 2021) | 44.1 | 68.2 | 77.0 | 24.7 | 45.1 | 54.6 |
| CLIP + GMAIL (ours) | **47.1** | **71.2** | **79.6** | **30.2** | **50.3** | **60.5** |
| Long-CLIP (Zhang et al., 2024) | 47.2 | 71.5 | 80.0 | 33.1 | 55.6 | 64.9 |
| Long-CLIP + GMAIL (ours) | **51.6** | **75.3** | **83.6** | **39.3** | **61.5** | **71.8** |

*Table 4.* **Zero-shot image classification.** We perform a zero-shot evaluation on pre-trained CLIP models for image classification on eight benchmarks. We report the top-1 accuracy to evaluate the quality of learned representations. The best results are indicated in **bold**.

| Method | DTD | Stanford Cars | SUN397 | Food 101 | Aircraft | Oxford Pets | Caltech 101 | ImageNet |
|---|---|---|---|---|---|---|---|---|
| CLIP (Radford et al., 2021) | 55.20 | 77.53 | 69.31 | 93.08 | 32.88 | 93.33 | 93.24 | 75.54 |
| CLIP + GMAIL (ours) | **65.26** | **81.32** | **75.53** | **95.21** | **37.85** | **95.23** | **95.57** | **77.68** |
| SynCLR (Tian et al., 2024) | 79.90 | 93.80 | 76.20 | 91.60 | 81.70 | 93.60 | 95.30 | 85.80 (ft) |
| SynCLR + GMAIL (ours) | **83.67** | **96.56** | **81.25** | **96.38** | **86.75** | **95.70** | **98.35** | **87.95 (ft)** |

*Table 5.* **Long caption retrieval on ShareGPT4V.** We report the image-to-text and text-to-image Recall@1 to evaluate the quality of retrieved images. The best results are indicated in **bold**.

| Method | Image-to-Text | Text-to-Image |
|---|---|---|
| CLIP (Radford et al., 2021) | 78.2 | 79.6 |
| CLIP + GMAIL (ours) | **85.2** | **86.7** |
| Long-CLIP (Zhang et al., 2024) | 94.6 | 93.3 |
| Long-CLIP + GMAIL (ours) | **97.2** | **96.1** |

Our CLIP + GMAIL approach achieves a top-1 accuracy of 65.26 on the DTD benchmark, outperforming the original CLIP (Radford et al., 2021) by 10.06 points, demonstrating the significant benefit of aligning generated images with real data. On the Stanford Cars dataset, our model reaches 81.32, showing robust performance gains, particularly in fine-grained classification tasks. For the challenging FGVC Aircraft benchmark, our method scores 37.85, marking a substantial improvement of 4.97 over the baseline CLIP, highlighting our model's capacity to handle complex visual distinctions. Additionally, our model performs exceptionally well on other benchmarks, achieving 75.53 on SUN397, 95.21 on Food 101, 95.23 on Oxford Pets, 95.57 on Caltech 101, and 77.68 on ImageNet 1K. These results consistently outperform both the standard CLIP and the CLIP + LoRA setup, confirming the effectiveness of our Gen-Real alignment strategy in broadening the model's generalization capabilities across various domains.

**Long Caption Retrieval.** We evaluate our model's capability to handle long captions using the ShareGPT4V (Chen et al., 2024) benchmark, as reported in Table 5. The evaluation focuses on image-to-text and text-to-image retrieval tasks, with Recall@1 used to assess the quality of retrieved results. Our model demonstrates an enhanced ability to comprehend and generate relevant responses to extended textual inputs, affirming its utility in applications that require detailed and descriptive outputs. For the CLIP-based models,

our CLIP + GMAIL configuration achieves 85.2 for image-to-text and 86.7 for text-to-image retrieval, outperforming both the original CLIP (Radford et al., 2021) and the CLIP + LoRA variants. This result highlights the effectiveness of our alignment strategy in bridging the semantic gap between generated and real images, particularly when handling complex, long-caption scenarios. When applied to the Long-CLIP architecture (Zhang et al., 2024), our Long-CLIP + GMAIL configuration reaches 97.2 for image-to-text and 96.1 for text-to-image retrieval, marking the highest performance among all tested configurations. These gains of 2.6 and 1.6 over Long-CLIP + LoRA confirm that our approach not only strengthens the alignment between modalities but also substantially improves the retrieval of images and captions involving extended and intricate descriptions. Overall, these results confirm the robustness and scalability of our framework in managing complex captioning tasks.

### 4.3. Experimental analysis

In this section, we performed ablation studies to demonstrate the benefit of Gen-Real alignment. We also conducted extensive experiments to explore the scaling trend on different training data sizes.

**Gen-Real Alignment.** To quantify the impact of Gen-Real alignment fine-tuning on our model's performance, we conducted ablation studies comparing models with and without alignment optimization. The results, presented in Table 6, demonstrate significant improvements across all metrics when alignment tuning is applied, validating the effectiveness of our proposed approach. In the context of image captioning tasks, models fine-tuned with Gen-Real alignment consistently outperform their counterparts that lack this optimization step. Specifically, adding Gen-Real alignment to the vanilla baseline using synthetic images to fine-tune all parameters led to substantial increases across all

*Table 6.* **Ablation study on Gen-Real Alignment.** We perform ablation studies on image captioning from pre-trained CLIP on generated images. The best results are indicated in **bold**.

| Alignment | B@4 (↑) | METEOR(↑) | CIDEr (↑) | SPICE (↑) | ROUGE-L (↑) | WMD (↑) |
|---|---|---|---|---|---|---|
| ✗ | 36.15 | 30.32 | 115.35 | 22.95 | 55.12 | 61.08 |
| ✓ | **38.12** | **31.67** | **119.53** | **23.75** | **56.27** | **62.16** |

*Table 7.* **Scaling trend of Gen-Real alignment on zero-shot image retrieval on Flickr30k.** We perform zero-shot retrieval on models trained from COCO, CC3M, and CC12M on Flickr30k. We report the Recall@1,5,10 metrics to evaluate the quality of retrieved images.

| Train Data | Image-to-Text | | | Text-to-Image | | |
|---|---|---|---|---|---|---|
| | R@1 (↑) | R@5 (↑) | R@10 (↑) | R@1 (↑) | R@5 (↑) | R@10 (↑) |
| COCO | 47.1 | 71.2 | 79.6 | 30.2 | 50.3 | 60.5 |
| CC3M | 48.6 | 73.6 | 82.2 | 32.6 | 52.6 | 62.3 |
| CC12M | **50.9** | **75.3** | **84.6** | **34.9** | **54.7** | **64.8** |

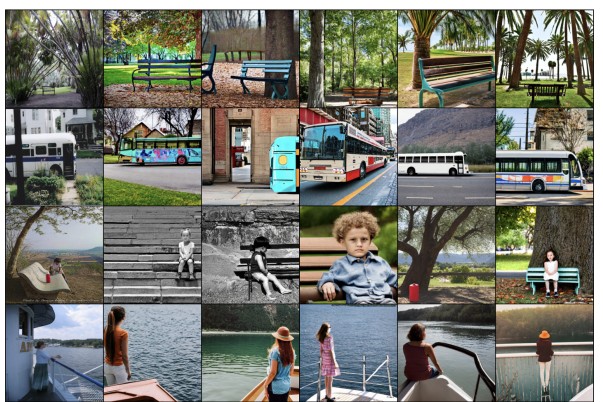

*Figure 2.* **Visualizations of real (Column 1) and generated images (Columns 2-6) using the same caption.** Those generated images generally capture high-level semantics in real images.

evaluated metrics: 3.56 in B@4, 1.13 in METEOR, 4.18 in CIDEr, 0.8 in SPICE, 1.15 in ROUGE-L, and 1.09 in WMD. These improvements highlight the critical role of alignment fine-tuning in bridging the modality gap between generated and real images, which enables the model to better capture and replicate the semantic richness found in real-world data. The results underscore the effectiveness of Gen-Real alignment in optimizing model performance, particularly in adapting to the nuances of generated images and their associated textual descriptions. By embedding generated images within the same latent space as real images, our approach enhances the model's ability to understand and process complex visual-language relationships, ultimately leading to superior performance in downstream tasks.

**Scaling trend of Gen-Real alignment.** To further evaluate the scalability of our proposed Gen-Real alignment, we explore its performance across varying scales of training data. Specifically, we apply our training framework on synthetic images derived from COCO (Lin et al., 2014), CC3M (Sharma et al., 2018), and CC12M (Changpinyo et al., 2021). The comparison results on zero-shot image retrieval on the Flickr30k benchmark are reported in Table 7.

*Table 8.* **Comparison with SigLIP on COCO captioning.** Our GMAIL significantly improves SigLIP by effectively addressing the synthetic-real discrepancy. The best results are **bold**.

| Method | B@4 (↑) | CIDEr (↑) |
|---|---|---|
| SigLIP | 37.51 | 117.82 |
| SigLIP + GMAIL (ours) | **42.35** | **125.68** |

The results reveal a clear scaling trend, where increasing the volume of training data from COCO to CC3M and then to CC12M consistently enhances the model's performance on both image-to-text and text-to-image retrieval tasks. Specifically, our model trained on CC12M achieves the highest scores with 50.9 R@1, 75.3 R@5, and 84.6 R@10 for image-to-text retrieval, and 34.9 R@1, 54.7 R@5, and 64.8 R@10 for text-to-image retrieval, outperforming the models trained on the smaller COCO and CC3M datasets. These improvements demonstrate that our Gen-Real alignment framework benefits significantly from larger and more diverse training datasets of generated images, effectively capturing richer semantic representations and enhancing retrieval capabilities. The results underscore the effectiveness of our method in leveraging the scaling trend of generated data, showing that as the scale of synthetic images increases, our model continues to learn and generalize better across zero-shot retrieval tasks.

**Visualization of Generated Images.** To further understand the quality and semantic alignment of the generated images used in our training process, we provide visualizations of a subset of generated images alongside their corresponding real counterparts, as shown in Figure 2. These images were generated using Stable Diffusion (Rombach et al., 2022), and are designed to closely match the real-world data in terms of visual realism and content. Through these visualizations, we observe that while generated images generally capture high-level features and structures present in real images, they may still exhibit subtle artifacts or variations that could contribute to the modality gap. Despite these differences, our Gen-Real Alignment framework successfully

*Table 9.* **Comparison with models trained on real images.** We perform experiments on image captioning from pre-trained CLIP on generated and real images. The best results are indicated in **bold**.

| Dual Projection | Alignment | Fine-tuning Data | B@4 (↑) | CIDEr (↑) | SPICE (↑) |
|:---:|:---:|:---:|:---:|:---:|:---:|
| ✗ | ✗ | ✗ | 32.15 | 108.35 | 20.12 |
| ✗ | ✗ | Synthetic | 36.15 | 115.35 | 22.95 |
| ✓ | ✓ | Synthetic | **38.12** | **119.53** | **23.75** |
| ✗ | ✗ | Real | 38.24 | 119.78 | 23.86 |
| ✓ | ✓ | Real | **38.37** | **119.95** | **23.98** |

*Table 10.* **Visual question answering on ScienceQA.** We report the average accuracy on questions with the image context from the ScienceQA benchmark. The best results are **bold**.

| Method | Accuracy (%) |
|:---|:---:|
| LLaVA | 85.2 |
| LLaVA + GMAIL (ours) | **87.6** |
| LLaMA-3 | 88.5 |
| LLaMA-3 + GMAIL (ours) | **91.2** |

*Table 11.* **Ablation study on LoRA rank and full fine-tuning.** We perform experiments on image captioning from pre-trained CLIP on generated images. The best results are indicated in **bold**.

| Method | B@4 (↑) | CIDEr (↑) | SPICE (↑) |
|:---|:---:|:---:|:---:|
| LoRA (rank = 2) | 36.85 | 117.62 | 23.10 |
| LoRA (rank = 4) | **38.12** | **119.53** | **23.75** |
| LoRA (rank = 6) | 37.96 | 119.12 | 23.60 |
| Full fine-tuning | 37.50 | 118.95 | 23.50 |

bridges this gap, as evidenced by the alignment of semantic features between the generated and real images in the learned latent space. The visualizations not only illustrate the potential of generated data as a cost-effective supplement to real-world data but also highlight the importance of explicit alignment strategies to mitigate discrepancies between generated and real data distributions.

**Comparison with SigLIP.** To strengthen the effectiveness of our work, we compare GMAIL with SigLIP (Zhai et al., 2023) on COCO captioning. The results are shown in Table 8. SigLIP (Zhai et al., 2023) adopts a sigmoid loss for better image-text pre-training, focusing solely on real images. In contrast, our GMAIL aligns real and generated images as distinct modalities, addressing the challenges of integrating synthetic data into further training. GMAIL is particularly relevant in scenarios requiring synthetic data, such as handling expensive attribute annotations or generating diverse samples. These results demonstrate that our GMAIL complements SigLIP by effectively addressing the synthetic-real discrepancy, allowing for enhanced generalization and performance improvements.

**Training on Real Images.** To illustrate the impact of real images on GMAIL, we compare performances on COCO captioning using real-only (first and fourth rows), mixed real-generated data (second row), and GMAIL alignment strategies (third and fifth rows). The results are shown in Table 9. These results indicate that GMAIL's alignment strategy not only bridges the synthetic-real gap but also could improve models trained exclusively on real data.

**ScienceQA Results.** We also evaluated GMAIL's performance on ScienceQA (Lu et al., 2022) when integrated with LLaVA (Liu et al., 2023) and LLaMA-3 (Meta, 2024). We calculated the average accuracy of questions with the image context. The comparison results are reported in Table 10.

These results highlight GMAIL's ability to enhance VLM's generalization across multimodal tasks.

**Ablation on LoRA.** LoRA (Hu et al., 2022) allows efficient adaptation to synthetic data while preserving the knowledge from pre-training on large-scale real data. This avoids the need for full fine-tuning, which can overwrite important pre-trained weights, especially when synthetic data is noisy or biased. The ablation results are reported in Table 11. As can be seen, LoRA with rank 4 achieves the best performance, balancing computational efficiency and alignment quality. Meanwhile, LoRA updates 35% fewer parameters compared to full fine-tuning while achieving better performance.

## 5. Conclusion

In this work, we present GMAIL, a novel framework for generative-to-real alignment that addresses the modality gap between generated and real images, a key challenge that often leads to mode collapse when integrating generated data into training pipelines. Our approach explicitly treats generated images as a separate modality and employs a training scheme that aligns these images within the same latent space as real images. By fine-tuning models on generated images, while maintaining a pre-trained model for real images, our framework facilitates explicit alignment between the two modalities, leading to significant performance improvements across various vision-language tasks. Extensive experiments demonstrate the efficacy of our method on a wide range of benchmarks, including image captioning, zero-shot image retrieval, and zero-shot image classification. Our results consistently show that GMAIL enhances the model's ability to generalize and perform across tasks, particularly when trained on large-scale datasets. The scaling trend observed with larger generated datasets such as CC12M further highlights the robustness and adaptability of our approach.

## Acknowledgements

This work was supported by the research fund of Hanyang University(HY-2024-2693) and the National Supercomputing Center with supercomputing resources including technical support(KSC-2024-CRE-0492).

## Impact Statement

Our work leverages generative models, such as stable diffusion models (Rombach et al., 2022), to create generated images that can supplement real-world datasets in training machine learning models. While this approach offers significant benefits in terms of reducing the need for expensive and time-consuming real-world data collection, we recognize the potential ethical risks associated with generated data. Generated images may inadvertently reflect biases present in the data used to train the generative models, potentially perpetuating harmful stereotypes or inaccuracies. To mitigate this, we emphasize the importance of careful curation of training datasets and encourage the community to develop strategies for auditing and debiasing generative models. Additionally, the alignment of generated data with real-world data must be handled with caution, as over-reliance on generated content can obscure important real-world variations.

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

# Appendix

In this appendix, we provide the following material:

- addition implementation and datasets details in Section A,

- algorithm for our GMAIL in Section B,

- more discussions on Gen-Real alignment in Section C,

- more experimental analyses in Section D,

- qualitative visualization results in Section E.

## A. Implementation & Dataset Details

In this section, we provide additional implementation details to ensure the reproducibility of our experiments, along with a comprehensive description of the datasets used.

**Implementation.** The base model used in our framework is the CLIP model (Radford et al., 2021), pre-trained on real images and paired with their textual descriptions. We fine-tune the pre-trained CLIP model on generated images using the LoRA (Hu et al., 2022) method to introduce low-rank updates, ensuring that the training remains computationally efficient. For contrastive learning, we set the temperature parameter $\tau = 0.07$ and optimize using the AdamW optimizer with a learning rate of $1 \times 10^{-4}$ and a batch size of 256. The synthetic training data were generated using Stable Diffusion v2 on NVIDIA A100-80GB GPUs. The number of generated images is consistent with the number of text-image pairs in the original training set: 560k for COCO, 3.3 million for CC3M, and 12 million for CC12M. Each image was generated with 50 inference steps, balancing quality and computational efficiency. The total generation time is 5 GPU days for COCO, 30 GPU days for CC3M, and 109 GPU days for CC12M. Parallelized generation was employed for larger datasets like CC12M. Fine-tuning for "Proj for Real" and "Proj for Gen" was performed for 50,000 steps.

**Datasets.** To evaluate the versatility and effectiveness of our Gen-Real Alignment framework, we employ a comprehensive suite of datasets across a variety of tasks, including image captioning, zero-shot image retrieval, and zero-shot image classification. This ensures a broad assessment of our model's performance across multiple domains and challenges.

- **COCO** (Lin et al., 2014): The COCO dataset is used for both image captioning and zero-shot image retrieval tasks. It offers a large and diverse collection of real-world images paired with detailed textual descriptions, serving as a benchmark for evaluating the alignment of generated and real image modalities.

- **Zero-Shot Image Classification**: To evaluate the generalization capabilities of our model, we utilize eight well-known benchmarks, following the setup of the original CLIP (Radford et al., 2021):

  - **DTD** (Cimpoi et al., 2014): Tests the model's ability to classify textures across various images.
  - **Stanford Cars** (Krause et al., 2013): A dataset focusing on fine-grained classification of car models, used to assess the model's capacity to distinguish between visually similar objects.
  - **SUN397** (Xiao et al., 2010; 2014): A large-scale scene classification dataset used to evaluate scene understanding.
  - **Food 101** (Bossard et al., 2014): A benchmark used to assess the model's ability to classify food items from various cuisines.
  - **Aircraft** (Maji et al., 2013): Used for fine-grained classification of aircraft models, testing the model's accuracy in distinguishing similar objects.
  - **Oxford Pets** (Parkhi et al., 2012): A dataset focused on the classification of various pet breeds, including both dogs and cats.
  - **Caltech 101** (Fei-Fei et al., 2004): A widely used object recognition dataset covering a variety of general categories.
  - **ImageNet 1K** (Deng et al., 2009): A benchmark for large-scale object classification, testing the model's ability to handle diverse image categories.

---

**Algorithm 1** GMAIL Algorithm: Training and Inference on Generated and Real Images

---

**Require:** Datasets of real images $\mathcal{D}_r = \{(x_r, y_r)\}$, generated images $\mathcal{D}_g = \{(x_g, y_g)\}$, and image pairs $\mathcal{D} = \{(x_g, x_r)\}$, pre-trained CLIP models $f_r$ and $f_g$, learning rate $\eta$, batch size $|\mathcal{B}|$, temperature $\tau$, LoRA parameters.

**Ensure:** Fine-tuned model $f_g$ for generated images, aligned with $f_r$ for real images.

1: **Initialize:** Load the pre-trained CLIP models $f_r$ and $f_g$ trained on real and generated images, respectively.
2: **Gen-CLIP Flow: Aligning Generated and Real Images.**
3: **for** each mini-batch $\mathcal{B} = \{(x_g, x_r)\}$ from $\mathcal{D}$ **do**
4:     Extract generated image features $f_g(x_g)$ for each $x_g \in \mathcal{B}$ using $f_g$.
5:     Extract real image features $f_r(x_r)$ for each $x_r \in \mathcal{B}$ using $f_r$.
6:     Compute cross-modality alignment loss $\mathcal{L}_{align}$:

$$\mathcal{L}_{align} = -\frac{1}{|\mathcal{B}|} \sum_{(x_g, x_r) \in \mathcal{B}} \log \frac{\exp(\text{sim}(f_g(x_g), f_r(x_r))/\tau)}{\sum_{x'_r \in \mathcal{B}} \exp(\text{sim}(f_g(x_g), f_r(x'_r))/\tau)}$$

7:     Apply LoRA updates to minimize $\mathcal{L}_{align}$.
8:     Update model parameters $f_g \leftarrow f_g - \eta \nabla_{f_g} \mathcal{L}_{align}$.
9: **end for**
10: **Alignment with Vision-Language Models for downstream tasks.**
11: **for** each LLM (e.g., from CLIPCap, LLaVA, LLaMA3) **do**
12:     Fine-tune the LLM using the aligned generated and real image embeddings from $\mathcal{D}_g$ and $\mathcal{D}_r$, respectively.
13: **end for**
14: **CLIP Flow: VLM Inference on Real Images.**
15: **for** each real image $x_r$ for inference **do**
16:     Extract real image features $f_r(x_r)$ using $f_r$.
17:     Use $f_r(x_r)$ instead $f_g(x_r)$ for inference the VLM model on real images.
18: **end for**
19: **Return:** VLM models with $f_g$ trained on generated images, aligned with the real-image model $f_r$.

---

- **CC3M** (Sharma et al., 2018) and **CC12M** (Changpinyo et al., 2021): These large-scale datasets provide millions of image-caption pairs, allowing us to explore the scalability of our Gen-Real alignment framework. We evaluate our model's performance when trained on both real and generated data from these expansive datasets.

- **ShareGPT4V**: To evaluate long caption retrieval, we use the ShareGPT4V dataset, which includes complex and descriptive captions associated with both generated and real images. This dataset emphasizes the importance of strong cross-modal alignment for retrieving long, detailed captions.

**Evaluation Metrics.** To comprehensively evaluate our framework, we employ task-specific metrics tailored to image captioning, zero-shot image retrieval, and zero-shot image classification:

- **Image Captioning**: Performance is assessed using standard metrics such as BLEU@4 (B@4) (Papineni et al., 2002), METEOR (Denkowski & Lavie, 2014), CIDEr (Vedantam et al., 2014), SPICE (Anderson et al., 2016), ROUGE-L (Lin & Och, 2004), and Word Mover's Distance (WMD) (Kusner et al., 2015). These metrics evaluate the quality and semantic accuracy of generated captions compared to the ground truth.

- **Zero-Shot Image Retrieval**: We measure both image-to-text and text-to-image retrieval capabilities using Recall@1, Recall@5, and Recall@10. These metrics assess the model's ability to correctly retrieve relevant items based on the provided query, highlighting its cross-modal understanding.

- **Zero-Shot Image Classification**: Classification performance on unseen categories is evaluated using top-1 accuracy, which reflects the model's generalization ability to classify new classes without prior training on those specific categories.

This experimental setup allows us to thoroughly validate our Gen-Real alignment framework across a wide range of

*Table 12.* **Computational costs comparisons on COCO training.** Our GMAIL introduces a slight increase in memory usage but remains more efficient on the convergence training time and steps than the baseline of indiscriminate mixing (gen+real) without alignment.

| Dual Projection | Alignment | Synthetic Data | Training Time (hrs) | Training Steps | Memory Usage (GB) | FLOPs (G) |
|---|---|---|---|---|---|---|
| ✗ | ✗ | ✗ | 8 | 50k | 24 | 70.2 |
| ✗ | ✗ | ✓ | 12 | 70k | 26 | 85.5 |
| ✓ | ✓ | ✓ | 10 | 60k | 28 | 85.5 |

*Table 13.* **Comparison with the same training steps.** We compare CLIP trained on real images, fine-tuned CLIP on generated images without alignment, and ours under COCO training with the same training steps. The best results are indicated in **bold**.

| Dual Projection | Alignment | Synthetic Data | B@4 ($\uparrow$) | CIDEr ($\uparrow$) | SPICE ($\uparrow$) |
|---|---|---|---|---|---|
| ✗ | ✗ | ✗ | 32.15 | 108.35 | 20.12 |
| ✗ | ✗ | ✓ | 35.76 | 113.42 | 22.63 |
| ✓ | ✓ | ✓ | **37.92** | **117.6** | **23.42** |

tasks, demonstrating its effectiveness in addressing the modality gap between generated and real images and enhancing performance across diverse vision-language applications.

## B. GMAIL Algorithm

In this section, we outline the algorithm that implements the Generative Modality Alignment for generated Image Learning (*GMAIL*) framework, incorporating the *Gen-CLIP* flow for training on generated images and the *CLIP* flow for inference on real images. This algorithm also details the cross-modality alignment loss and how we ensure alignment with vision-language models (VLMs) such as CLIPCap (Mokady et al., 2021), LLaVA (Liu et al., 2023), and LLaMA-3 (Meta, 2024).

Algorithm 1 summarizes the training and inference process for the GMAIL framework, detailing how the model is trained on generated images using the *Gen-CLIP* flow, and subsequently applied to real images during inference. The algorithm also explains how to integrate aligned generated and real data with vision-language models such as CLIPCap, LLaVA, and LLaMA-3 for downstream tasks.

## C. More Discussions on Gen-Real Alignment

In this section, we provide a comprehensive discussion of Gen-Real Alignment. Given training samples having the same text: real image $R$, synthetic $S$, and text $T$, let us denote our dual encoders as $f, g, h$ for real-encoder, syn-encoder, and text-encoder, respectively.

**Single vs. Dual Modality.** In a single-modality scenario (*i.e.*, a single encoder setup where $f = g$), given training would reduce distance $D(f(R), h(T))$ and $D(f(S), h(T))$, and then $D(f(R), f(S))$ would be reduced together. However, due to the nature of synthetic images, there could exist a gap between $R$ and $S$, such as unnatural artifacts, assuming $S$ contains spurious features. Therefore, under such approaches to put real and generated images into the same embedding space, generated artifacts may dominate, causing poor generalization and overfitting to synthetic patterns. Moreover, if the encoder ignores such different inputs $R$ and $S$, and produces representations that remain constant and equal, it can lead to "mode collapse" (LeCun, 2022; Assran et al., 2023), where the model overfits generated patterns, degrading performance on real data. On this line, we consider a dual-modality scenario, (*i.e.*, dual encoder setup where $f \neq g$) to prevent such a problem caused by reducing a distance $D(f(R), f(S))$. Here, we instead minimize $D(f(R), h(T))$ and $D(g(S), h(T))$, so allowing a small $D(f(R), h(S))$, not $D(f(R), f(S))$. Specifically, the expected role of $h$ is to ignore a synthetic complement of $S$ and produce representations that remain an intersection of $S$ and $R$ (having the same $T$). Such separate mappings of $f$ and $g$ would allow learning focused on shared characteristics between the real and generated modalities. Thereby treating generated images as a distinct modality, GMAIL could prevent "mode collapse", enabling the effective use of synthetic data to augment real datasets without poor generalization and overfitting to synthetic patterns.

**Cross-Modality Alignment Loss.** Furthermore, the proposed cross-modality alignment loss aims to directly reduce a distance $D(f(R), h(S))$ allowing effective and faster training to convergence. As shown in Table 12, the proposed loss

*Table 14.* **Quantitative similarity metrics comparisons on COCO.** We computed the cosine similarity between paired real and generated image embeddings without and with alignment on COCO dataset.

| Alignment | Cosine Similarity ($\uparrow$) |
|:---:|:---:|
| ✗ | 0.52 |
| ✓ | 0.89 |

*Table 15.* **Multi-modal reasoning and visual-language alignment on MMMU.** We evaluated GMAIL's performance on MMMU benchmark when integrated with LLaVA. We report the average accuracy on questions with the image context. The best results are **bold**.

| Method | Accuracy (%) |
|:---|:---:|
| LLaVA | 44.7 |
| LLaVA + GMAIL (ours) | **48.3** |

reduced training time and steps to convergence. Throughout our extensive experiments, for a given $R$ and $S$ having the same $T$, we have demonstrated the effect of minimizing a distance $D(f(R), h(S))$ which learns shared semantics between real and generated images while ignoring generated artifacts of $S$ may raise poor generalization on real images.

**Empirical Validation of Alignment Loss.** Nevertheless, we further conducted an ablation study on the effect of the cross-modality alignment loss (*i.e.*, the effects of directly reducing $D(f(R), h(S))$) under the dual encoder setup on COCO captioning. The results in Table 6 confirm that the alignment loss significantly bridges the modality gap, resulting in consistent performance improvements.

## D. More Experimental Analysis

**Computational Costs.** We performed additional experiments to compare the computational costs. Table 12 shows the results, including explicit details on the contributions of the cross-modality alignment loss and dual-model setup. The additional costs for GMAIL stem from the cross-modality alignment loss, which facilitates aligning the features of generated and real images in a shared latent space, and the dual-projection setup, which processes the two modalities separately. Compared to CLIP without the dual projection, our GMAIL introduces a slight increase in memory usage but remains more efficient on the convergence training time and steps than the baseline of indiscriminate mixing on generative and real data without the alignment.

**Same Training Steps.** Here, we have conducted additional experiments with equal training steps for CLIPCap (Mokady et al., 2021) across all compared methods and included the results in the Table 13. Even with identical training steps, GMAIL consistently outperforms both baselines, confirming that gains stem from our alignment design, not from additional training.

**MMMU Results.** We also evaluated GMAIL's performance on MMMU (Yue et al., 2024), which is a benchmark that requires multi-modal reasoning and visual-language alignment. Table 15 shows that GMAIL can improve visual grounding and alignment with LLMs, demonstrating generalization to reasoning-based VLM tasks.

**Comparison with Tasks2Sim.** Here, we have added experiments comparing Tasks2Sim (Mishra et al., 2022) in Table 16. Specifically, GMAIL outperforms prior mix-only methods due to explicit modality disentanglement and alignment, rather than blending synthetic and real data blindly.

**Other generative models.** We have conducted experiments using FLUX (Labs, 2024), which introduces a more powerful and differently parameterized generation pipeline compared to Stable Diffusion v2. These additional results in Table 17 allow us to test GMAIL's robustness to shifts in generator-specific artifacts. The performance improvements remain consistent with FLUX, indicating robust alignment across varying artifact styles and photorealism levels.

**Qualitative Embeddings Visualization.** To further validate the alignment between real and generated data, we conducted t-SNE (van der Maaten & Hinton, 2008) visualizations and cosine similarity analyses of the embeddings without and with alignment. Figure 3 shows the t-SNE plots of real and generated embeddings from 1000 samples in the COCO dataset. Without alignment, real and synthetic embeddings form two distinct clusters, reflecting the modality gap. With alignment proposed in our GMAIL, the gap between real and synthetic embeddings is significantly reduced, with both modalities aligning closely.

*Table 16.* **Comparison to Task2Sim.** We report a comparison to Task2Sim (Mishra et al., 2022), which introduced a mixed training approach (joint usage of real and synthetic data). The best results are indicated in **bold**.

| Method | B@4 (↑) | CIDEr (↑) | SPICE (↑) |
|---|---|---|---|
| Task2Sim | 36.65 | 115.72 | 23.02 |
| GMAIL (ours) | **38.12** | **119.53** | **23.75** |

*Table 17.* **Image generation with FLUX.** We replace a generative model from Stable Diffusion v2 to FLUX and perform experiments on the image captioning task. The best results are indicated in **bold**.

| Method | B@4 (↑) | CIDEr (↑) | SPICE (↑) |
|---|---|---|---|
| FLUX (without alignment) | 37.20 | 117.82 | 23.40 |
| FLUX + GMAIL (ours) | **39.54** | **122.36** | **24.15** |

**Quantitative Similarity Metrics.** We also quantified the alignment score of the above 1000 samples in the COCO dataset using cosine similarity between paired real and generated embeddings. The results are shown in Table 14. These results demonstrate that the alignment loss effectively bridges the Gen-Real gap, ensuring better feature consistency across modalities. Moreover, without alignment, we observed that the cosine similarity between embeddings of real image-text pairs and generated image-text pairs was only 0.44 and 0.42, respectively. This empirically demonstrates that the misalignment between real and generated images is comparable to the gap between images and text embeddings of CLIP models.

## E. Qualitative Visualizations

In this section, we provide qualitative visualizations of the generated images used in our experiments. Figures 4, 5, 6, 7, 8 and 9 show examples of images generated by Stable Diffusion (Rombach et al., 2022), alongside their corresponding real-world counterparts from the COCO dataset (Lin et al., 2014). Our visualizations demonstrate that the generated images closely resemble real images, capturing key semantic details and structural elements. However, subtle differences in texture or object placement are occasionally present. These artifacts highlight the importance of our Gen-Real Alignment (GMAIL) framework, which ensures that these differences do not lead to mode collapse by aligning the feature representations of generated and real images in the latent space. These visualizations further validate the effectiveness of our alignment strategy, ensuring that both generated and real data contribute equally to the model's understanding during inference.

w/o alignment

w alignment

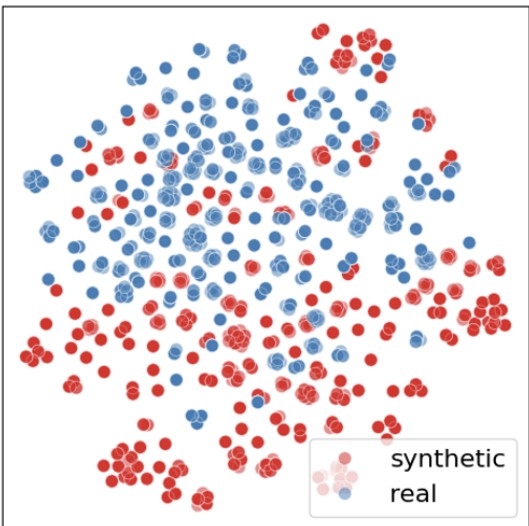

*Figure 3.* **Qualitative Visualizations of embeddings of real and synthetic images without (Left) and with (Right) alignment.** Blue and red dots denote the embeddings for real and synthetic images, respectively. Our GMAIL with alignment significantly reduced the gap between real and synthetic images, with both modalities aligning closely in the latent space.

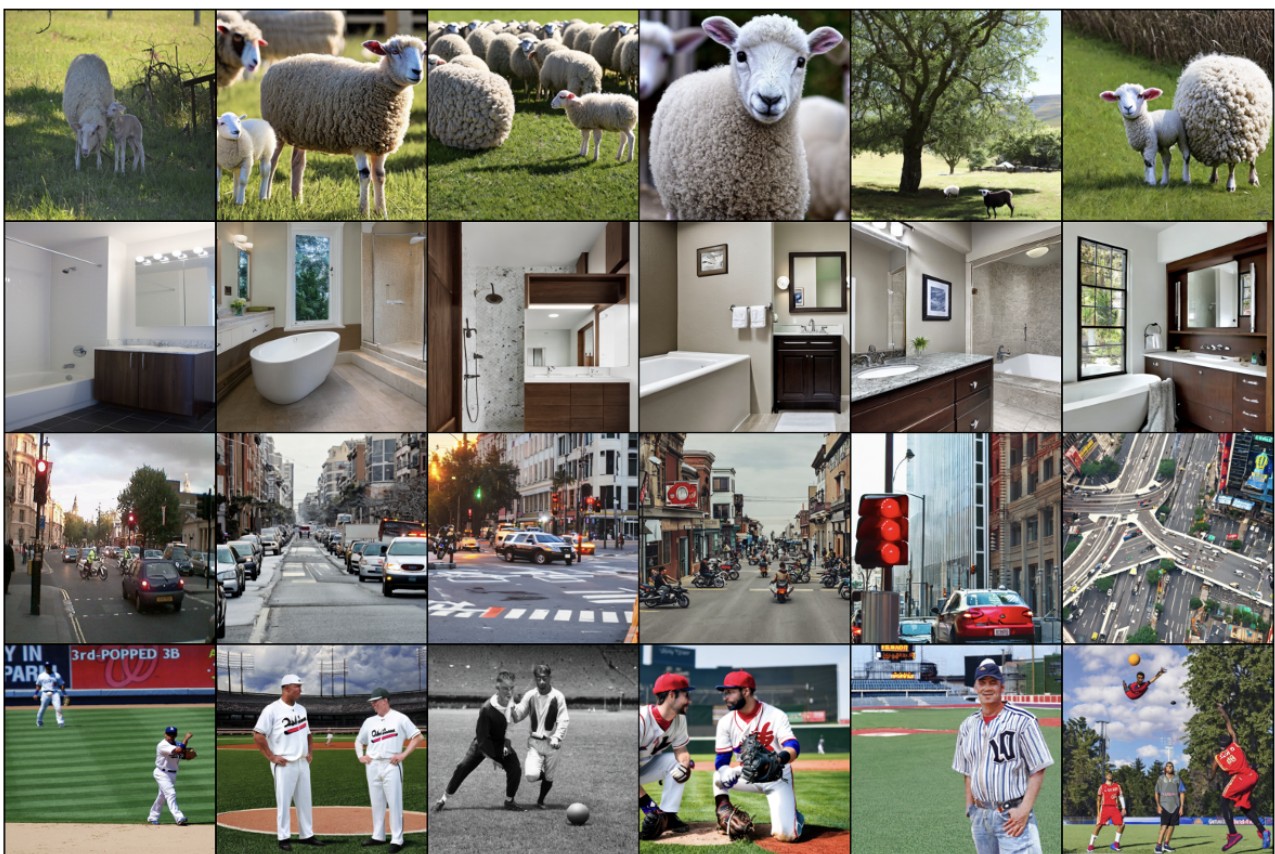

*Figure 4.* **Visualizations of real (Column 1) and generated images (Columns 2-6) using the same caption.** Those generated images generally capture high-level semantics in real images.

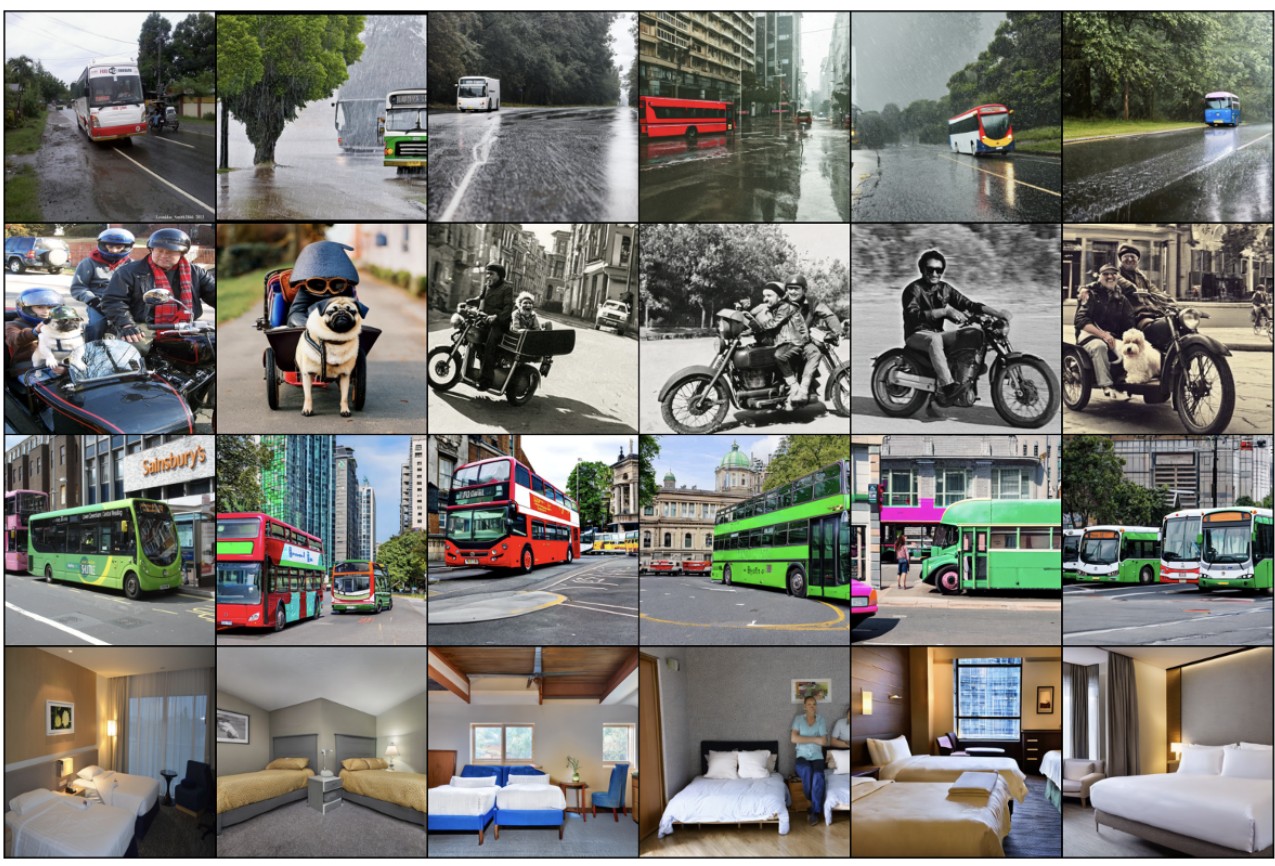

*Figure 5.* **Visualizations of real (Column 1) and generated images (Columns 2-6) using the same caption.** Those generated images generally capture high-level semantics in real images.

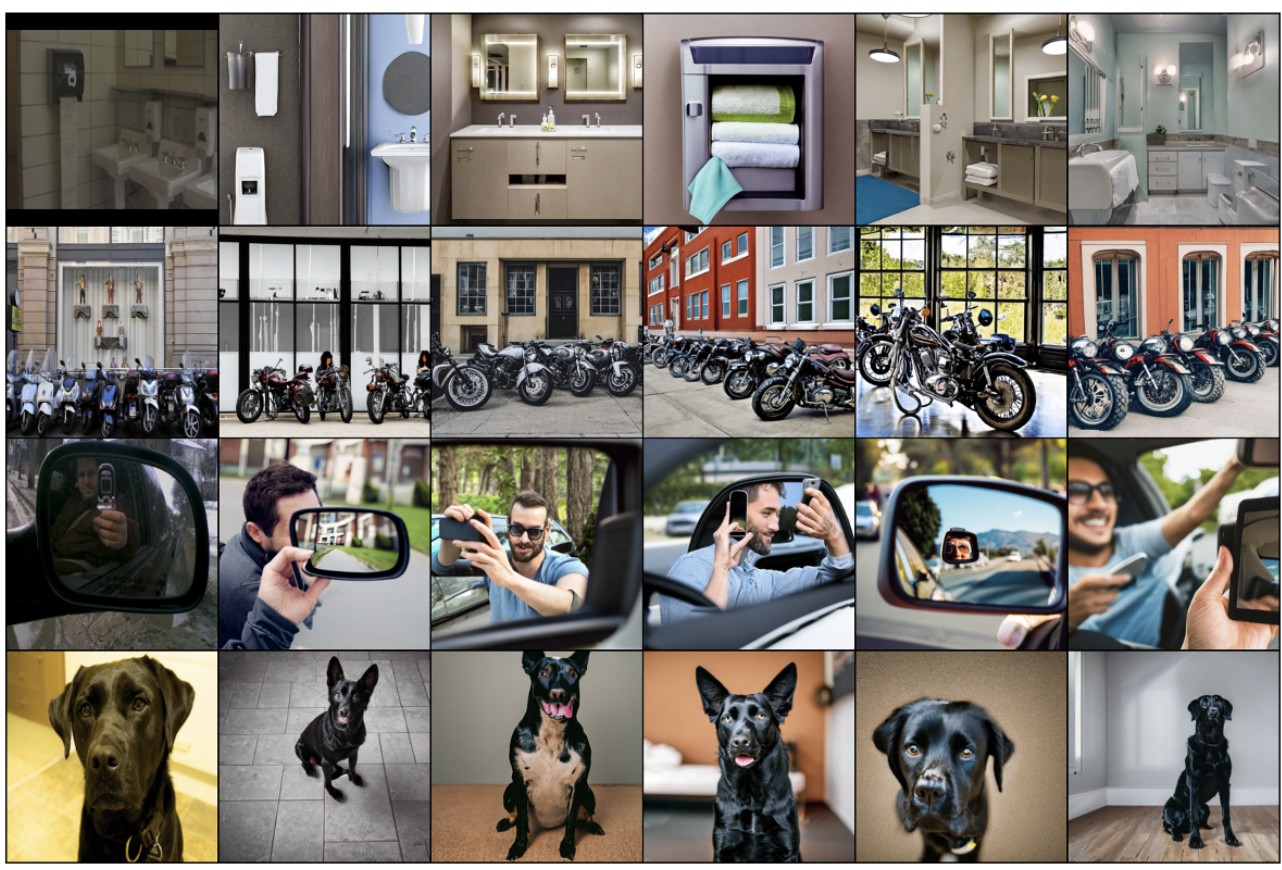

*Figure 6.* **Visualizations of real (Column 1) and generated images (Columns 2-6) using the same caption.** Those generated images generally capture high-level semantics in real images.

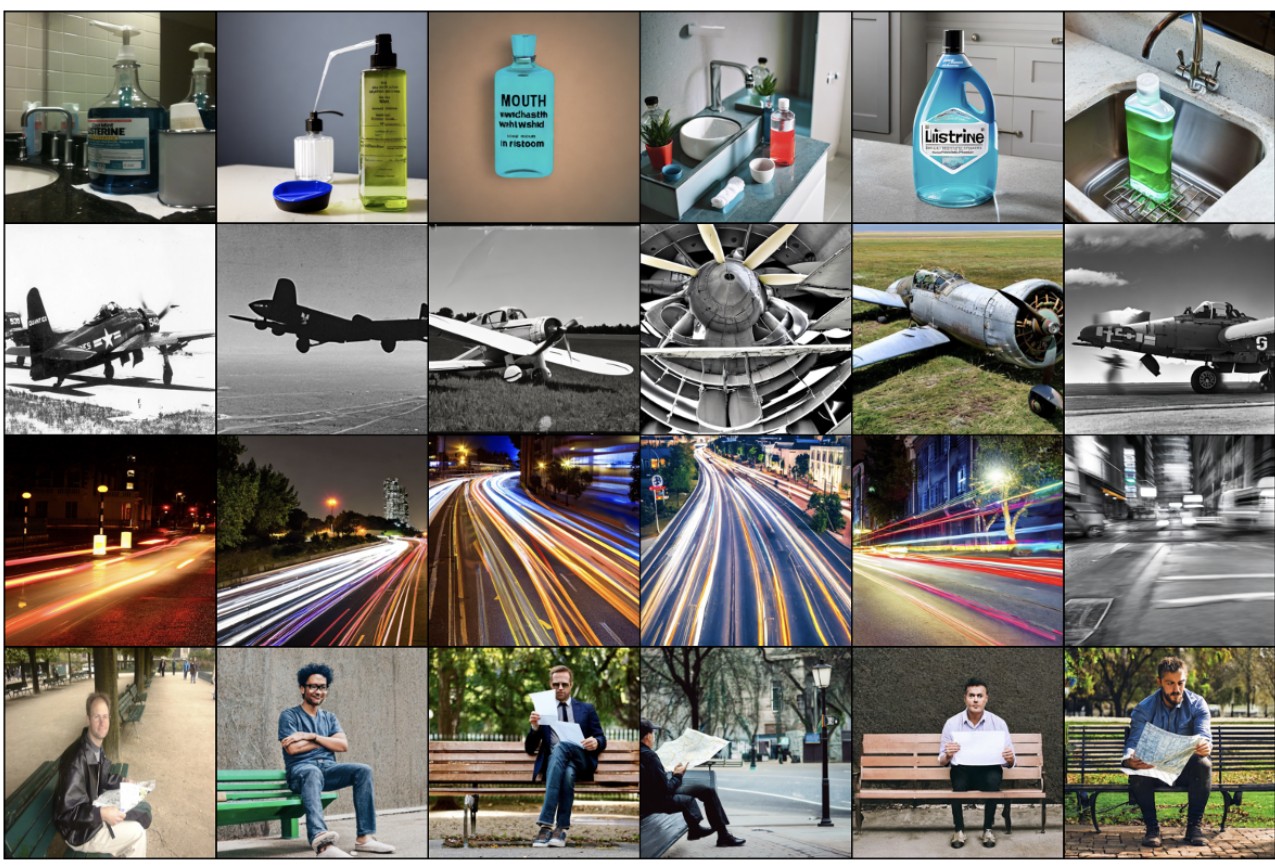

Figure 7. **Visualizations of real (Column 1) and generated images (Columns 2-6) using the same caption.** Those generated images generally capture high-level semantics in real images.

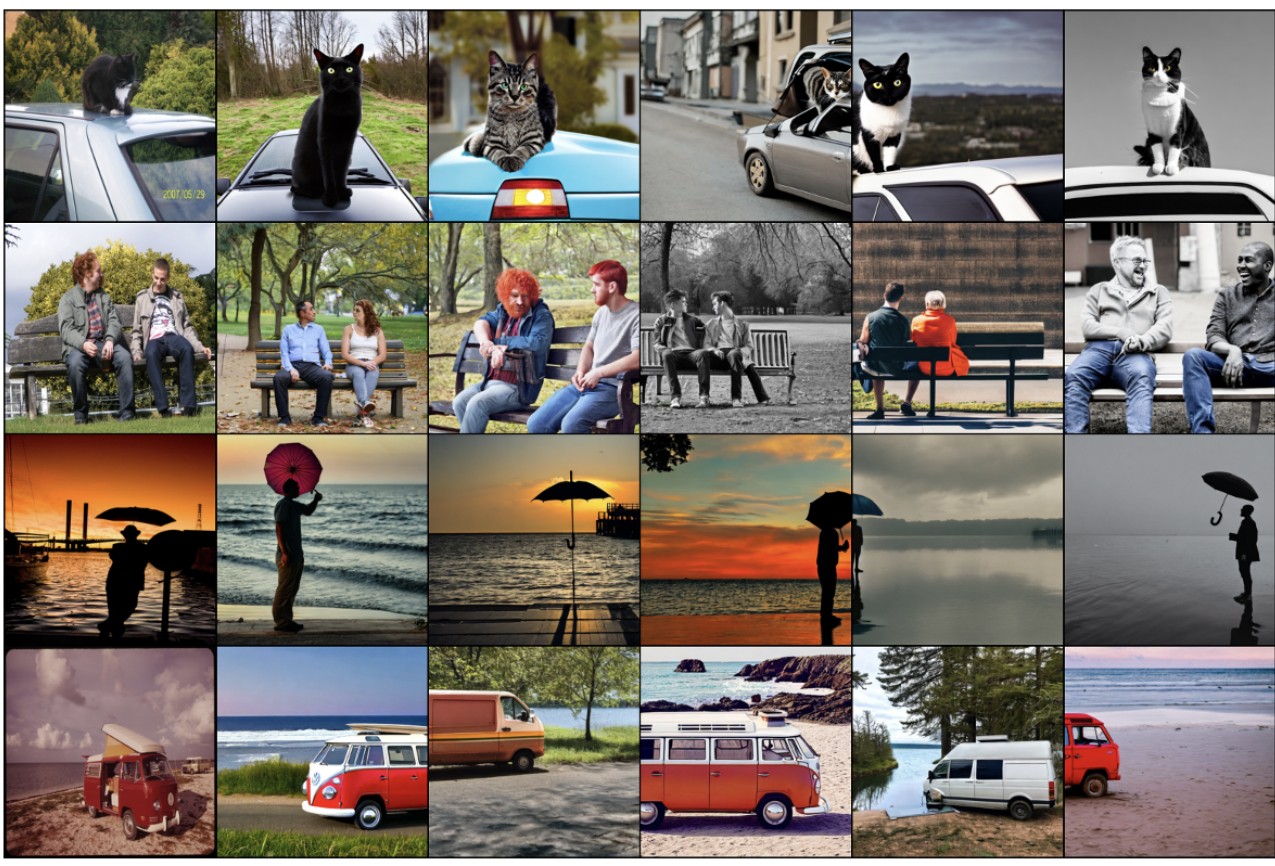

*Figure 8.* **Visualizations of real (Column 1) and generated images (Columns 2-6) using the same caption.** Those generated images generally capture high-level semantics in real images.

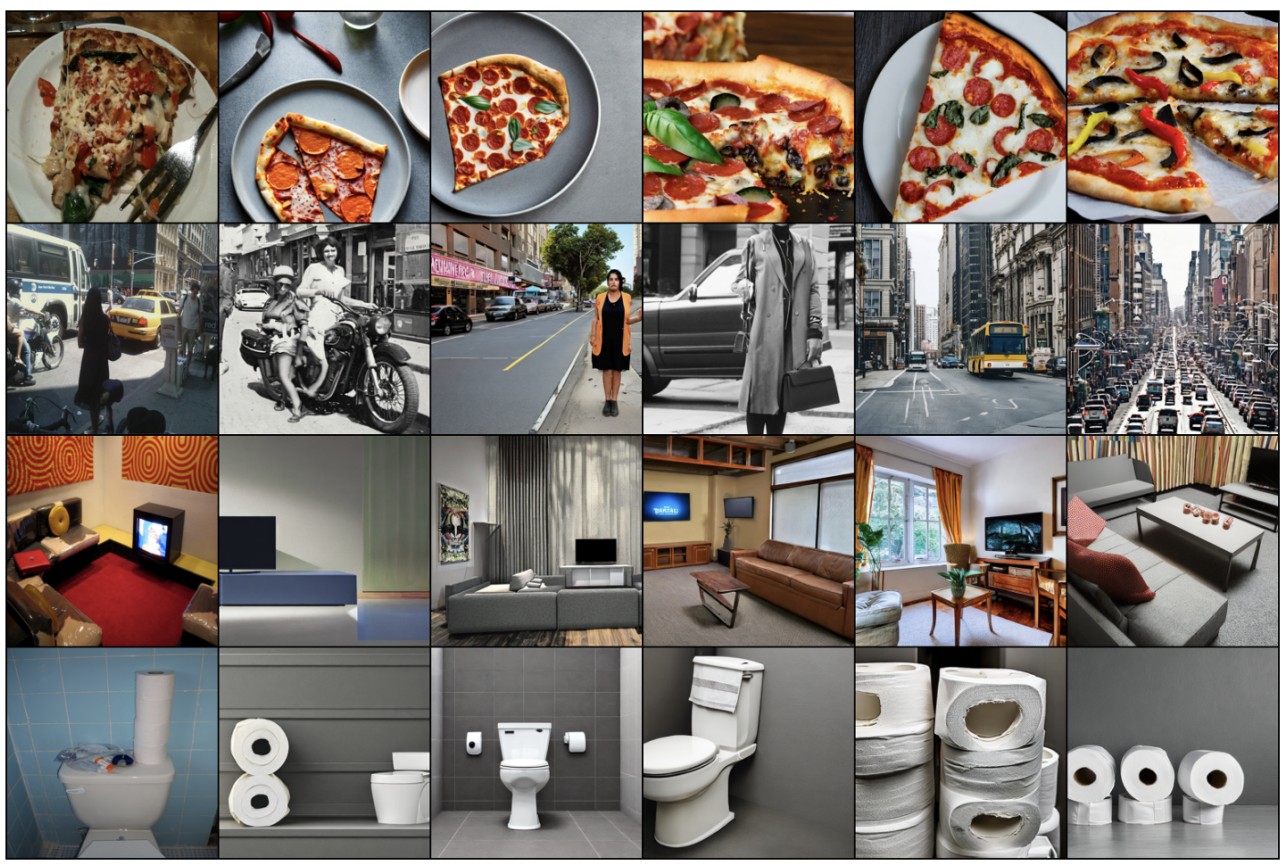

*Figure 9.* **Visualizations of real (Column 1) and generated images (Columns 2-6) using the same caption.** Those generated images generally capture high-level semantics in real images.

