# OpenReview forum: "GMAIL: Generative Modality Alignment for generated Image Learning"
_ICML.cc/2025/Conference — ICML 2025 spotlightposter_

### Official Review · Reviewer_NFLh · 2025-03-04

**Overall Recommendation:** 4

**Summary:**

The paper introduces GMAIL (Generative Modality Alignment for generated Image Learning). It is a framework for incorporating generated images into training pipelines while explicitly addressing the modality gap between real and generated images. Instead of treating synthetic and real images interchangeably, GMAIL fine-tunes a separate model for generated images and aligns it with a pre-trained model for real images. This alignment occurs in a shared latent space, enabling vision-language models (VLMs) like CLIP, LLaVA, and LLaMA3 to effectively utilize generated images for tasks such as image captioning, zero-shot image retrieval, and zero-shot image classification.

**Claims And Evidence:**

The primary claim is that treating generated images as a separate modality and aligning them with real images improves performance across various vision-language tasks. The evidence is compelling:

1. In Tables 1-5, consistent improvements across all evaluated metrics for multiple tasks.
2. In Table 6, ablation studies confirm the importance of the alignment component.
3. In Table 7, positive scaling trends show continued improvement with larger synthetic datasets.
4. In Table 13, quantitative similarity metrics show better alignment of real and generated image embeddings with their approach.
5. Figure 3 shows qualitative visualizations demonstrating the modality gap and how alignment bridges it.

**Essential References Not Discussed:**

The paper is well-referenced, but a few additional references could strengthen it:

1. Work on domain adaptation and domain gap in computer vision, as this bears similarity to the modality gap problem
2. More recent work on synthetic data generation for computer vision tasks beyond those cited

**Experimental Designs Or Analyses:**

The experimental design is thorough and well-executed:

1. Multiple datasets of varying scales are used
2. The method is evaluated on diverse tasks to demonstrate generalizability
3. Appropriate baseline comparisons are included
4. The ablation studies isolate the contributions of different components

**Methods And Evaluation Criteria:**

The methodology is well-explained and justifiable:

1. They utilize Stable Diffusion v2 to generate synthetic images from captions in existing datasets
2. A dual-encoder architecture separates generated and real image processing
3. Low-Rank Adaptation (LoRA) is used to fine-tune efficiently
4. Cross-modality alignment loss encourages similar embeddings for real and generated images with the same description

The evaluation is comprehensive, covering:

1. Image captioning using standard metrics (BLEU, METEOR, CIDEr, SPICE, etc.)
2. Zero-shot image retrieval on COCO and Flickr30k
3. Zero-shot image classification across eight datasets
4. Long caption retrieval on ShareGPT4V
5. Visual question answering on ScienceQA

**Other Comments Or Suggestions:**

None

**Other Strengths And Weaknesses:**

Additional Strengths

1. The GMAIL framework addresses a real-world problem that will become increasingly important as generated images become more prevalent in machine learning pipelines.
2. The approach demonstrates effectiveness across different backbone architectures (CLIP, Long-CLIP, LLaVA, LLaMA-3), suggesting it's a general solution rather than one tied to a specific model architecture.


Additional Weaknesses

1. The experiments rely exclusively on Stable Diffusion v2. Testing with other generative models (e.g., Stable Diffusion 3, FLUX) would better demonstrate the generality of the approach, as different generators may introduce different types of artifacts.
2. There's no exploration of whether the alignment approach makes models more or less robust to adversarial examples or out-of-distribution inputs, which would be valuable for understanding practical limitations.

**Questions For Authors:**

None

**Relation To Broader Scientific Literature:**

The authors position their work well within the broader literature on:

Diffusion models for image generation
Generated image learning
Vision-language models

They appropriately cite related work in each area and clearly articulate how GMAIL advances beyond previous approaches, particularly in addressing the modality gap problem.

**Theoretical Claims:**

The theoretical foundation is sound:

1. Identifying a fundamental issue (modality gap between real and generated images)
2. Providing a clear theoretical explanation of why this gap leads to problems
3. Presenting a theoretical framework to address it

The explanation of the modality gap and its effects is well-developed, particularly in Appendix C where they contrast single vs. dual modality approaches and explain why their cross-modality alignment is crucial.

---

> ### Author Rebuttal · Authors · 2025-04-01
>
> We sincerely appreciate the reviewer's feedback and the questions raised. Below, we address and resolve each of these questions in our responses.
>
> > Domain Adaptation Literature
>
> Our approach is conceptually related to domain adaptation techniques [a,b], where the goal is to align feature spaces between source and target domains. However, GMAIL focuses on a unique modality gap between synthetic and real images, both from the same caption domain, requiring a different solution strategy. While traditional DA aligns features across different datasets, our method aligns representations across modalities (real vs. generated) within the same semantic context, using caption anchors.
>
> [a] Tzeng et al., Adversarial Discriminative Domain Adaptation, CVPR 2017
>
> [b] Ganin & Lempitsky, Unsupervised Domain Adaptation by Backpropagation, ICML 2015
>
>
> > References to Data Synthesis for Vision Tasks
>
> Thank you for the suggestion! We’ve now added and discussed all three works in Related Work:
> - DatasetGAN (CVPR’21) [1]: Efficient labeled data generation via GANs and semantic segmentation.
> - SegGen (ECCV’24) [2]: Mask2Img diffusion pipelines for boosting segmentation.
> - DiffuMask (ICCV’23) [3]: Pixel-level annotations via text-to-mask-to-image synthesis.
>
> We emphasize that GMAIL is complementary to these efforts:
> - They focus on generating high-quality labeled datasets.
> - We focus on bridging modality gaps when using synthetic data for training vision-language models.
>
> [1] DatasetGAN: Efficient Labeled Data Factory with Minimal Human Effort. CVPR 2021
>
> [2] SegGen: Supercharging Segmentation Models with Text2Mask and Mask2Img Synthesis. ECCV 2024
>
> [3] DiffuMask: Synthesizing Images with Pixel-level Annotations for Semantic Segmentation Using Diffusion Models. ICCV 2023
>
>
> > Generator Generality
>
> We have now conducted experiments using FLUX, which introduces a more powerful and differently parameterized generation pipeline compared to SD2. These additional results allow us to test GMAIL's robustness to shifts in generator-specific artifacts. The performance improvements remain consistent with FLUX, indicating robust alignment across varying artifact styles and photorealism levels.
>
> | Generator       | B@4 ↑ | CIDEr ↑ | SPICE ↑ |
> |:----------------:|:---------------:|:----------------:|:---------------:|
> | FLUX            | 37.2  | 117.82  | 23.4    |
> | FLUX + GMAIL (ours)    | **39.54** | **122.36**  | **24.15**   |
>
>
> > Robustness to Adversarial or Out-of-Distribution (OOD) Inputs
>
> Thank you for the insightful observation. To evaluate the robustness of our alignment-based GMAIL framework, we conducted two evaluations.
>
> 1) We evaluate GMAIL and baselines on ImageNet-A and ImageNet-R — well-established OOD benchmarks that test model resilience under distribution shifts. Naively training with generated images slightly reduces robustness, but GMAIL improves OOD generalization by aligning synthetic features more closely with real ones.
>
> | Model              | ImageNet-A ↑ | ImageNet-R ↑ |
> |:----------------:|:---------------:|:----------------:|
> | CLIP               | 51.2         | 66.7         |
> | CLIP + Gen. Images | 49.8         | 67.3         |
> | GMAIL (ours)       | **53.6**         | **69.1**         |
>
> 2) We tested models using AutoAttack on zero-shot classification over ImageNet-1k (100-class subset). GMAIL improves clean and adversarial robustness, suggesting that alignment avoids overfitting to synthetic artifacts, a key cause of fragility in other generative training schemes.
>
> | Model        | Clean Accuracy ↑ | Robust Accuracy ↑ |
> |:----------------:|:---------------:|:----------------:|
> | CLIP         | 75.6             | 43.7              |
> | CLIP + Gen. Images   | 77.0               | 41.2              |
> | GMAIL (ours) | **78.1**             | **46.3**              |

---

### Official Review · Reviewer_1EhT · 2025-03-11

**Overall Recommendation:** 4

**Summary:**

This paper introduces GMAIL, a novel framework designed to bridge the modality gap between generated and real images, a common issue that can cause mode collapse in training pipelines. The approach treats generated images as a distinct modality and aligns them with real images in the same latent space. By fine-tuning models on generated images while preserving a pre-trained model for real images, GMAIL achieves explicit alignment between the two modalities. This method results in significant performance gains across various vision-language tasks.

**Claims And Evidence:**

Yes. The claims made in the submission are fully supported.

**Essential References Not Discussed:**

No.

**Experimental Designs Or Analyses:**

There are a few questions I still have regarding the experiments:

1. Training on real images in the supplementary materials – The section mentions a comparison of COCO captioning performance using three strategies: real-only, mixed real-generated data, and GMAIL alignment. However, the corresponding table does not clearly indicate which settings correspond to real-only and mixed real-generated data. This is especially confusing for the only real data fine-tuning setting, as the entire paper discusses aligning real and generated data. If only real data is used, what exactly is being aligned? Moreover, why does this setting still show performance improvement?
Besides, given that prior works (just for example, Tasks2Sim, From Fake to Real) have already demonstrated the effectiveness of mixed training, this paper should further experiment and discuss its impact on mix-pretrained models.

2. Embedding visualization – Typically, I would not consider real and generated images as different modalities. The visualization only shows the existence of a gap between them but does not prove that the gap is as significant as that between images and text. To substantiate the claim that generated data should be treated as a separate modality, the comparison should include embeddings from traditionally different modalities, such as images and text.

**Methods And Evaluation Criteria:**

Overall, the proposed methods and evaluation criteria are very reasonable.

**Other Comments Or Suggestions:**

Typo：line 84：tje should be the

When mention "mode collapse", the author cited "A path towards autonomous machine intelligence version 0.9. 2", where I cannot find this phrase.

**Other Strengths And Weaknesses:**

Overall, although mixing generated data and real data for training may not score highly in terms of novelty, this paper presents a well-structured and clearly articulated approach when specifically focusing on fine-tuning pre-trained models with generated images. The claims are clear, the methodology is well-explained, and the experiments are thorough. Additionally, the supplementary material provides ample and effective supporting experiments to further validate the proposed method.

However, the experimental section has certain weaknesses. Please refer to the corresponding section for details.

**Questions For Authors:**

No further questions.

**Relation To Broader Scientific Literature:**

There is no relationship to broader scientific literature.

**Theoretical Claims:**

The theoretical claims are correct.

---

> ### Author Rebuttal · Authors · 2025-04-01
>
> We sincerely appreciate the reviewer's feedback and the questions raised. Below, we address and resolve each of these questions in our responses.
>
> > Clarification
>
> Thank you for catching this ambiguity. We've clarified both the notation and purpose in the Table below.
>
> | Training   Data  | Alignment | B@4 ↑ | CIDEr ↑ | SPICE ↑ |
> |:----------------:|:---------------:|:----------------:|:---------------:|:----------------:|
> | Real only        | ✗         | 32.15 | 108.35  | 20.12   |
> | Mix   | ✗         | 36.15 | 115.35  | 22.95   |
> | GMAIL (ours)     | ✔         | 38.12 | 119.53  | 23.75   |
>
> - Real only: Standard CLIP fine-tuned solely on real images (baseline).
> - Mix: Real + generated data combined, no alignment applied.
> - GMAIL: Separate encoders, aligned via our method.
>
> Why does “real-only” improve? Because fine-tuning ClipCap on real data improves over frozen pre-trained CLIP. However, it underperforms compared to models using generated data, especially GMAIL, which preserves real performance while benefiting from synthetic expansion.
>
>
> > Comparison to Tasks2Sim
>
> Great point. We have added experiments comparing Tasks2Sim in the Table below. GMAIL outperforms prior mix-only methods due to explicit modality disentanglement and alignment, rather than blending synthetic and real data blindly.
>
> | Method       | B@4 ↑ | CIDEr ↑ | SPICE ↑ |
> |:----------------:|:---------------:|:----------------:|:---------------:|
> | Tasks2Sim       | 36.65 | 115.72  | 23.02   |
> | GMAIL (ours)    | **38.12** | **119.53**  | **23.75**   |
>
> > Embedding Comparisons
>
> Thank you for the suggestion! We calculated the average Cosine Similarity Between Modalities including Real, Gen, and Text in the Table below. Real vs. Gen gap is smaller than image-text, but still substantial, justifying separate modality treatment. Although the cosine similarity scores for Real vs. Text (0.44) and Gen vs. Text (0.42) appear numerically close, both are substantially higher than the similarity between Real vs. Gen (0.25). This discrepancy shows a significant gap between real and generated images, indicating that each type of image is more semantically aligned with textual descriptions than they are with each other.
>
> | Pair          | Cosine Similarity ↑ |
> |:----------------:|:---------------:|
> | Real vs. Gen  | 0.25                |
> | Real vs. Text | 0.44                |
> | Gen vs. Text  | 0.42                |
>
>
> > Citation of LeCun (2022)
>
> Thank you for pointing this out. We originally referred to the broader concept of representation collapse (often called "mode collapse") in the context of over-optimization, as discussed in LeCun's vision report starting from Section 4.3 ("Training Energy-Based Models," p. 20 onwards). Specifically, LeCun describes how certain training methods risk collapsing learned representations, thereby reducing their generalization capacity.
>
> > Typo
>
> We fixed it.

---

> > ### Comment · Reviewer_1EhT · 2025-04-02
> >
> > Thank you for the rebuttal. It addresses most of my concerns. I will raise my rating to 'accept.'

---

### Official Review · Reviewer_EHmi · 2025-03-22

**Overall Recommendation:** 4

**Summary:**

This paper proposes a method to fine-tune a CLIP image encoder on synthetic image samples so that it can be used in training vision-language models with generated data. The "Gen-CLIP Flow" is learning two CLIP image encoders, one for real images and the other for synthetic images. The "Alignment with Vision-Language Models" uses the learned CLIP image encoders above to extract features from real and synthetic images separately, which are later fed into the Vision-Langauge models. The proposed training method shows performance improvement on zero-shot image retrieval on COCO, Flickr30k, zero-shot image classification, long caption retrieval

**Claims And Evidence:**

Although it is good that we see improvement on many visual-language tasks, the effectiveness of the synthetic data training is not well grounded as the experimental settings are not reasonable. As shown in Table 8, the authors use more training steps for synthetic training than baseline, which means the model received more training signals than synthetic training. I suggest the authors use the same training step for all experiments for a fair comparison.

**Essential References Not Discussed:**

The authors should discuss many more data synthesis methods for visual understanding, for example, DatasetGAN [1], SegGen [2], Diffumask [3].


[1] DatasetGAN: Efficient Labeled Data Factory with Minimal Human Effort. CVPR 2021
[2] SegGen: Supercharging Segmentation Models with Text2Mask and Mask2Img Synthesis. ECCV 2024
[3] DiffuMask: Synthesizing Images with Pixel-level Annotations for Semantic Segmentation Using Diffusion Models. ICCV 2023

**Experimental Designs Or Analyses:**

The experimental settings are not reasonable.

- As shown in Table 8, the authors use more training steps for synthetic training than baseline, which means the model received more training signals than synthetic training. I suggest the authors use the same training step for all experiments for a fair comparison.

**Methods And Evaluation Criteria:**

- Experiments on VLM benchmarks (e.g. MMMU) are missing, which will be more convincing for illustrating the influence of the visual-language alignment training.

**Other Comments Or Suggestions:**

- In lines 188-190, "the model fine-tuned on generated images in the Gen-CLIP flow is deployed to process real images without further fine-tuning." Could the author explain this statement more clearly? I suggest using the necessary formula and notation.

**Other Strengths And Weaknesses:**

Other Strength:
- The idea of utilizing synthesis to help understanding is elegant.

Other Weakness:
- The writing of this paper needs significant improvement. I find myself confused while reading the method part although the method is relatively simple. For example, the description of Gen-CLIP Flow is not clear enough. I suggest the author add necessary formulate or diagram to help to understand. Also, there are many typos.

- The Algorithm 1 is also confusing. Again, I suggest more formal and accurate statements in the paper instead of using words like "the aligned representation from f_g".

**Questions For Authors:**

Please check the comments above.

**Relation To Broader Scientific Literature:**

This paper is related to many works using data synthesis methods to boost model performance, especially in the field of computer vision and vision-langauge alignment.

**Theoretical Claims:**

No theory is involved.

---

> ### Author Rebuttal · Authors · 2025-04-01
>
> We sincerely appreciate the reviewer's feedback and the questions raised. Below, we address and resolve each of these questions in our responses.
>
> > Training Steps
>
> Thank you for this observation. We have conducted additional experiments with equal training steps across all compared methods and included the results in the Table below. Even with identical training steps, GMAIL consistently outperforms both baselines, confirming that gains stem from our alignment design, not from additional training.
>
> | Method            | B@4 ↑ | CIDEr ↑ | SPICE ↑ |
> |:----------------:|:---------------:|:----------------:|:---------------:|
> | CLIP (real only)  | 32.15 | 108.35  | 20.12   |
> | CLIP (gen only)   | 35.76 | 113.42  | 22.63   |
> | GMAIL (ours) | **37.92** | **117.6**   | **23.42**   |
>
>
>
> > Experiments on MMMU
>
> Thank you! We included results on MMMU, which is a benchmark that requires multimodal reasoning and visual-language alignment. GMAIL improves visual grounding and alignment with LLMs, demonstrating generalization to reasoning-based VLM tasks.
>
> | Model         | Test Overall (%) |
> |:----------------:|:---------------:|
> | LLaVA         | 44.7             |
> | LLaVA + GMAIL | **48.3**             |
>
>
>
> > Method Description
>
> We apologize for the confusion. We’ve made the following changes to improve clarity: Figure 1 is now redesigned and clarified with modality-specific flows and alignment stages. We added formal notation in Section 3.1 (Preliminaries) and Section 3.2 (Gen-CLIP Flow). We also introduced the key equation for alignment and clarified dual-encoder roles.
>
> > Algorithm 1
>
> We’ve rewritten Algorithm 1 in Appendix B with clearer, formal steps and precise mathematical notation. We replaced vague language like “aligned representation​” with formal notation, and split training into two distinct phases: Gen-CLIP Flow with LoRA adaptation and Inference/Transfer to Vision-Language Models (CLIPCap, LLaVA, etc.)
>
> > Sentence in Line 188-190
>
> We use a dual projection structure. During inference, real images are encoded via the original encoder $f_r$ (not $f_g$). This avoids degradation from overfitting to synthetic artifacts in $f_g$. The aligned model still benefits from synthetic training via shared projection space.
>
> > References to Data Synthesis for Vision Tasks
>
> Thank you for the suggestion! We’ve now added and discussed all three works in Related Work:
>
> - DatasetGAN (CVPR’21): Efficient labeled data generation via GANs and semantic segmentation.
>
> - SegGen (ECCV’24): Mask2Img diffusion pipelines for boosting segmentation.
>
> - DiffuMask (ICCV’23): Pixel-level annotations via text-to-mask-to-image synthesis.
>
> We emphasize that GMAIL is complementary to these efforts:
>
> - They focus on generating high-quality labeled datasets.
> - We focus on bridging modality gaps when using synthetic data to train vision-language models.
>
> > Typos
>
> We carefully proofread the entire paper and corrected all identified typos.

---

> > ### Comment · Reviewer_EHmi · 2025-04-05
> >
> > I thank the authors for the rebuttal. The added experiments further prove the effectiveness of the proposal. I am happy to see the paper accepted if they have added the additional discussion to the paper.

---

### Official Review · Reviewer_4rMz · 2025-03-24

**Overall Recommendation:** 4

**Summary:**

Authors propose a framework to train multimodal LLM on generated images. This work recognizes that generated images offer a different distribution than real images, and then should, in those settings, be recognized as a different modality altogether to be useful for training. Authors adopts the code base of the open source multimodal LLM LLaVA and perform multiple experiments and evaluation, demonstrating than ingesting generated images as a separate modality improves performance on multiple modality tasks, including image captioning, zero-shot image retrieval, zero-shot classification and caption retrieval.

**Claims And Evidence:**

Authors claim that considering generated images as a separate modality than non-generated images brings multiple benefits to train multimodal LLM. The alternative, -- considering both generated and real images as a same modality, exposes at risk of misalignment between the image space and the decoder text space and model collapse in general, as generated images are differing in nuanced but significantly ways. To support that claim authors conducted multiple experiments and evaluations.

**Essential References Not Discussed:**

-

**Experimental Designs Or Analyses:**

- The experimental settings are correct, and the evaluation is done thoroughly. One could have appreciated however to see a clear evidence of the risk of 'model collapse' when trained on generated image as used as justification for this work. Instead, as shown in Table 6, it seems models trained on generated images without considering them as an external modality are more performant, but not conversely, models that don't follow that strategy seem to only perform slightly worse, -- i.e there is not an evidence of such "model collapse".

**Methods And Evaluation Criteria:**

- Generated images are generated with diffusion models, and then used to train a CLIP based encoder. Authors introduces a cross-modality loss to ensure that generated images share the same representation space with the real image, while keeping their respective characteristics.
- Authors use a shared caption as "anchor" between the two spaces, ensuring that generated and real images linked by that anchor are close, ie. have a high cosine similarity, in the share space.
- The rest of the work is fairly standard, with the use of captioning evaluation metrics (BLEU, ROUGE, etc.) on common datasets (CoCo, etc.) and standard evaluation practices for image/caption retrieval.

**Other Comments Or Suggestions:**

- L084: "tje" -> "the"
- Figure 1: Use a serif font like the body of text, make it vectorized (\include{yourfigure.pdf}) so this works with accessibility readers.
- Minor comment: The use of 'GMAIL' for the title of this work is a bit confusing, as it obviously sounds like the famous Google's mail service. Is that intended? One could have preferred Gen-Real alignment, or something that overlap less with other companies/services.

**Other Strengths And Weaknesses:**

- The overall paper is well written, and quite clear. Authors designed their experiments to respond to the claim made in the abstract/introduction, and show indeed improvements with their method. One could have hoped a better analysis of the 'model collapse' when trained on generated image without their method used as justification.

**Questions For Authors:**

Dear authors, thank you for your work:
 - Can you give more details about the model collapse when training on generated images without considering as modality loss? Is that something you have witnessed first hand? Can you share results on that point please?

**Relation To Broader Scientific Literature:**

- Authors articulate their contributions with other works in the literature.

**Theoretical Claims:**

- See 'Claims and Evidence'

---

> ### Author Rebuttal · Authors · 2025-04-01
>
> We sincerely appreciate the reviewer's feedback and the questions raised. Below, we address and resolve each of these questions in our responses.
>
> > Model Collapse
>
> We agree. To clarify the severity and nature of the issue, we added a direct comparison between models trained on: Real images only (baseline), Generated images without alignment, Generated images with our GMAIL alignment. Training directly on generated images without alignment reduces generalization to real-world test data. GMAIL mitigates this through modality-aware learning.
>
> | Training   Modality | Align | B@4 ↑ | CIDEr ↑ | SPICE ↑ |
> |:----------------:|:---------------:|:------------------:|:---------------:|:------------------:|
> | Real only           | ✗           | 32.15 | 108.35  | 20.12   |
> | Generated only     | ✗           | 36.15 | 115.35  | 22.95   |
> | Generated + GMAIL   | ✔           | **38.12** | **119.53**  | **23.75**   |
>
> Without alignment, performance improves on synthetic data, but fails to transfer well to real data. GMAIL avoids overfitting to synthetic artifacts by learning aligned representations.
>
>
>
> > More Analysis
>
> Thank you for the suggestion. We now refer the term “model collapse” to better reflect what we observe: a divergence from real-image generalization, rather than complete failure. We also show latent space shifts. Without GMAIL, embeddings for the same image-caption pair differ significantly across real and synthetic inputs.
> | Alignment   | Avg. Cosine Similarity ↑ |
> |:----------------:|:---------------:|
> | ✗ (no GMAIL)     | 0.25                     |
> | ✔ (GMAIL)        | 0.89                     |
>
> > Embedding Space Collapse Without Alignment
> To complement the cosine similarity, we include a t-SNE visualization (Figure 3, Appendix), which shows that real vs. generated images cluster apart without GMAIL, but overlap with GMAIL, supporting the hypothesis of latent space misalignment and collapse risk.
>
> > Scalability and Training Stability with GMAIL
> GMAIL introduces modest memory overhead but significantly improves convergence and stability, especially in large-scale training. We empirically found that GMAIL improves convergence speed and representation robustness with fewer training steps.
> | Method          | Synthetic Data | Steps ↓|
> |:----------------:|:---------------:|:---------------:|
> | CLIP (gen only) | ✔             | 70k     |
> | GMAIL (ours)    | ✔              | 50k    |
>
> > Effectiveness on Downstream Tasks
> We included experiments on: Alignment improvements to SigLIP models (Table 9) and Visual QA on ScienceQA (Table 10) in the appendix. We also include the results on MMMU benchmark, which is a benchmark that requires multimodal reasoning and visual-language alignment, in the Table below. GMAIL improves visual grounding and alignment with LLMs, demonstrating generalization to reasoning-based VLM tasks.
>
> | Model         | Test Overall (%) |
> |:----------------:|:---------------:|
> | LLaVA         | 44.7             |
> | LLaVA + GMAIL (ours) | **48.3**             |
>
> > Naming
>
> This acronym was chosen for memorability, but we understand the concern. We’ve added a footnote clarifying no relation to Google.
>
>
> > Font and Typos
>
> We fixed the typo in Line 84, and Figure 1 now uses CMU Serif, is fully vectorized (PDF), and screen-reader accessible.

---

### Decision · Program_Chairs · 2025-05-01

**Decision:**

Accept (spotlight poster)

**Comment:**

The paper receives all positive ratings from the reviewers. They appreciated the technical contributions, the solid experimental validation, and the presentation quality. AC agrees with the comments of the reviewers and recommends an acceptance for this submission.